# MITIGATING EMERGENT ROBUSTNESS DEGRADATION WHILE SCALING GRAPH LEARNING

**Xiangchi Yuan**[1*]    **Chunhui Zhang**[2*]    **Yijun Tian**[3]    **Yanfang Ye**[3]    **Chuxu Zhang**[1]

[1]Brandeis University, {`xiangchiyuan,chuxuzhang`}`@brandeis.edu`
[2]Dartmouth College, {`chunhui.zhang.gr`}`@dartmouth.edu`
[3]University of Notre Dame, {`yijun.tian,yye7`}`@nd.edu`

## ABSTRACT

While graph neural networks have exhibited remarkable performance in various graph tasks, a significant concern is their vulnerability to adversarial attacks. Consequently, many defense methods have been proposed to alleviate the deleterious effects of adversarial attacks and learn robust graph representations. However, most of them are difficult to *simultaneously* avoid two major limitations: 1) an emergent and severe degradation in robustness when exposed to very intense attacks, and 2) heavy computation complexity hinders them from scaling to large graphs. In response to these challenges, we introduce an innovative graph defense method for unpredictable real-world scenarios by *designing a graph robust learning framework that is resistant to robustness degradation* and *refraining from the unscalable designs with heavy computation*: specifically, our method employs a denoising module, which eliminates edges that are associated with attacked nodes to reconstruct a cleaner graph; Then, it applies Mixture-of-Experts to select differentially private noises with varying magnitudes to counteract the hidden features attacked at different intensities toward robust predictions; Moreover, our overall design avoids the reliance on heavy adjacency matrix computations, such as SVD, thus facilitating its applicability even on large graphs. Comprehensive experiments have been conducted to demonstrate the anti-degraded robustness and scalability of our method, as compared to popular graph adversarial learning methods, under diverse attack intensities and various datasets of different sizes.

## 1 INTRODUCTION

Graph Neural Networks (GNNs) are potent tools for learning relational data, excelling in tasks like node and graph classification (Kipf & Welling, 2017; Veličković et al., 2018; Hamilton et al., 2017; Ying et al., 2018; You et al., 2020; Guo et al., 2023b). They employ message-passing to iteratively update node representations by aggregating neighbor information. GNNs find widespread use, such as social media (Wu et al., 2022; Zhang et al., 2022; Qian et al., 2022; Wen et al., 2022b; Li et al., 2023a), recommender systems (Huang et al., 2021; Tian et al., 2022; Fan et al., 2019; Ouyang et al., 2024, 2023), and molecular prediction (Jin et al., 2018; Guo et al., 2023a; Wang et al., 2023).

While GNNs have excelled in various tasks, they are vulnerable to adversarial attacks (Zügner & Günnemann, 2019; Zheng et al., 2021; Tian et al., 2023b; Zügner et al., 2018) that manipulate their performance through tactics like edge modifications (Geisler et al., 2021), node perturbations (Zügner & Günnemann, 2019; Sun et al., 2020), or malicious node injections (Zou et al., 2021). To address these vulnerabilities, several defense mechanisms have emerged, but they often face two significant challenges: 1) severe robustness degradation: many existing defense methods, such as RGCN (Zhu et al., 2019) and GNN-SVD (Entezari et al., 2020), effectively handle mild node injection attacks but experience an emergent decline in robustness as attack intensity increases. As shown in Figure 1, when attack intensity surpasses a threshold of 300 injected nodes, error rates for many models surge by more than 50%. This limits their suitability for real-world scenarios (Goodfellow et al., 2015; Zheng et al., 2021; Madry et al., 2018; Zou et al., 2021). 2) limited scalability: scalability is a notable concern, particularly for widely used methods like GNN-SVD (Entezari et al., 2020) and

---

*Xiangchi is mentored by Chunhui; both contributed equally and are listed alphabetically by last name.

GNNGuard (Zhang & Zitnik, 2020), which require dense adjacency matrix computations. This can result in substantial computational overhead, as shown in the experiment section, leading to out-of-memory problems when applied to larger datasets like Flick, Reddit, and AMiner, especially when using a 32 GB GPU. These challenges necessitate innovative solutions to enhance the robustness and scalability of GNNs against adversarial attacks.

Therefore, we propose a novel framework called **DRAGON** (i.e., **D**ifferentially **P**rivate **M**asked **G**raph Aut**o**-E**n**coder for Anti-degraded Robustness): To address (i) the emergent robustness degradation under increasing-intensity graph attacks, DRAGON utilizes a denoise masked auto-encoder to reconstruct the given attacked graph towards cleaner node connections, and further uses Mixture-of-Experts Liu et al. (2023); Zhang et al. (2023c), which is specifically differential privacy-based, to eliminate the impact of injected nodes hidden in an attacked graph. For dealing with (ii) the limited scalability issue on large graph datasets, DRAGON avoids the heavy computation of large adjacency matrices, which is achieved by making all designs not require large-scale adjacency matrices like GNNSVD (Entezari et al., 2020) and GNNGuard (Zhang & Zitnik, 2020), thereby we prevent out-of-memory from occurring in our framework while scaling our framework to large graph datasets.

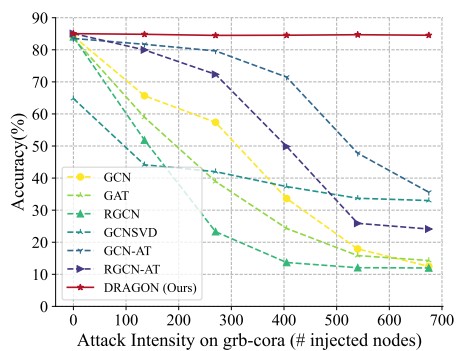

Figure 1: The performance of different methods when FGSM attack intensity on grb-Cora is increasing. DRAGON outperforms other robust baselines with negligible robustness degradation even under extreme-intensity attack.

The above effects are delivered in two steps: *First*, given an attack graph as input, a masked graph auto-encoder is applied to eliminate malicious edges connected to injected nodes, resulting in a (nearly) clean graph with very few injected nodes connected to clean nodes; *Second*, we ensemble the differential privacy (DP) mechanism in each GNN layer, which introduces random noises into the graph features while counteracting the attacked node features and constraining the change of the model's output, thus against attack perturbations in injected nodes. Additionally, the Mixture-of-Experts (MoEs) technique is used to manipulate multiple DP expert networks, each holding Gaussian DP noise of different magnitudes. These expert networks are then assigned to injected node features attacked at different intensities, providing an appropriate level of noise to improve robustness. Our major contributions are summarized as follows:

- Our solution, DRAGON framework, addresses both severe robustness degradation and scalability constraints in adversarial graph learning. Existing techniques often struggle with GNN scalability on large datasets and maintaining robustness against various attack intensities.

- Our framework comprises two key components: a denoise auto-encoder for cleaning attacked graphs by removing malicious edges connected to injected nodes and a differentially-private MoE that adapts the level of differential privacy noise to counteract attacks of different intensities.

- We extensively test DRAGON on diverse graph datasets and varying-intensity attacks, where it outperforms popular baselines, ensuring robustness in unpredictable real-world scenarios while preventing emergent phenomena – severe robustness degradation.

## 2 RELATED WORK

**Graph Neural Networks.** GNNs distinguish in various graph mining tasks (Hamilton et al., 2017; Battaglia et al., 2018; Xu et al., 2019; Zhang et al., 2023b,a; Jia et al., 2024) due to their prowess in learning non-Euclidean data. Early GNN variants like GCN (Kipf & Welling, 2017; Gao et al., 2018; Wu et al., 2019a) introduced convolutional concepts to graph data. Subsequently, graph attention networks(Veličković et al., 2018; Wang et al., 2019) incorporated attention mechanisms. Moreover, masked autoencoders (He et al., 2022) found application in the graph domain (Li et al., 2023b; Tian et al., 2023a), inspiring our work in constructing clean graphs from attacked ones.

**Adversarial Learning on Graphs.** Graph attacks disrupt structural or semantic graph properties via connection manipulation (Du et al., 2017; Chen et al., 2018; Waniek et al., 2018), node feature perturbation (Zügner et al., 2018; Zügner & Günnemann, 2019; Sun et al., 2020; Zhang et al., 2023c;

Jia et al., 2023), or malicious node insertion (Wang et al., 2020; Zou et al., 2021). In response to these attacks, researchers have proposed a range of defense methods for GNNs. These include learning robust GNNs (Zhu et al., 2019; Feng et al., 2020; Jin et al., 2020), adversarial training (Yue et al., 2022; Wen et al., 2022a; Zhang et al., 2023c), and removing attacked inputs during preprocessing (Entezari et al., 2020; Zhang & Zitnik, 2020). However, two critical challenges persist: severe robustness degradation under intense attacks and limited scalability on large graph datasets. These inspire us to design the DRAGON framework to uniquely address both challenges.

## 3 PRELIMINARIES

**Sparse Mixture-of-Experts.** The Mixture-of-Experts (MoE) combines multiple models to make predictions by partitioning the input space and assigning experts to these partitions. Each expert specializes in a specific input subset, and their predictions are weighted and assembled using a gating mechanism. Given input space $h$ and an MoE layer with $N$ experts $\mathbf{E} = \{E_i(\cdot)\}_{i=1}^{N}$, the MoE output is: $y = \sum_{i=1}^{N} p_i(h) E_i(h)$, where $E_i$ is an expert model, $p_i(h)$ is the expert's weight, typically determined by a gating network.

**Node Injection Attacks.** In node injection attacks, malicious nodes are inserted into an undirected attributed graph $\mathcal{G} = (\mathcal{V}, \mathcal{E}, \mathcal{X})$. As GRB benchmark (Zheng et al., 2021) formally defined, the goal is to maximize the number of error predictions without modifying existing edges or node features:

$$\max_{\mathcal{G}'} |\arg\max f(\mathcal{G}') \neq \arg\max f(\mathcal{G})|, \tag{1}$$

where $\mathcal{G}' = (\mathcal{V}', \mathcal{E}', \mathcal{X}')$, and $\mathcal{V}'$ includes malicious nodes. Constraints on $\mathcal{E}'$ and $\mathcal{X}'$ limit edge changes and node feature modifications too.

**Differential Privacy.** Differential Privacy (DP) protects individual data during computations by adding randomness. It ensures that the output of a function $f(\cdot)$ on neighboring datasets is indistinguishable. Formally, a mechanism $\mathcal{M}$ is $(\epsilon, \delta)$-differentially private if, for neighboring datasets $D \sim D'$ and events in the output space $\mathcal{O}$:

$$\Pr[\mathcal{M}(D) \in \mathcal{O}] \leq e^{\epsilon} \Pr[\mathcal{M}(D') \in \mathcal{O}] + \delta, \tag{2}$$

with parameters $\epsilon$ and $\delta$ controlling privacy strength and budget. For any $\sigma, \delta \in (0, 1)$, the deviation of the classical Gaussian DP mechanism has the form $\sigma = \sqrt{2 \ln(1.25/\delta)}/\epsilon$ (Dwork et al., 2006; Balle & Wang, 2018; Dwork et al., 2014). DP introduces noise to the function output, maintaining output consistency despite input perturbations, aligning with adversarial robustness in deep learning.

## 4 METHODOLOGY

We introduce DRAGON for enhancing GNN robustness. Figure 2 depicts the framework's two key components: first, DMGAN serves as an attacked graph preprocessing module, removing malicious edges linked to injected nodes, thus restoring a cleaner graph for the defender GNN; second, the framework combines the DP mechanism with GConv to form a DP-GConv layer, enhancing the defense module. Given unpredictable attack intensities, we partition DP-GConv into multiple experts, each with varying DP noise magnitudes, enabling defense against attacks of varying strengths.

### 4.1 DENOISE MASKED GRAPH AUTO-ENCODER

Denoise Masked Graph Auto-Encoder (DMGAN) eliminates the negative influence of injected nodes by removing the malicious edges associated with the injected nodes. And its pipeline can be divided into three steps: masking the input (attacked) graph, encoding and decoding, and self-supervising.

**Mask the Input Graph.** Given an input graph $\mathcal{G}$, we first adopt the widely used random walk (Perozzi et al., 2014) as a path-masking strategy to mask the graph. We denote the masked subgraph as $\mathcal{G}_{\mathrm{m}} = (\mathcal{V}_{\mathrm{m}}, \mathcal{E}_{\mathrm{m}}, \mathcal{X}_{\mathrm{m}})$, and the visible subgraph as $\mathcal{G}_{\mathrm{v}} = (\mathcal{V}_{\mathrm{v}}, \mathcal{E}_{\mathrm{v}}, \mathcal{X}_{\mathrm{v}})$, where they complement each other, i.e., $\mathcal{G}_{\mathrm{m}} \cup \mathcal{G}_{\mathrm{v}} = \mathcal{G}$. To illustrate, the path masking strategy selects and masks multiple adjacent edges via random walk. This process is formulated as follows: $\mathcal{E}_{\mathrm{m}} \sim \mathrm{RandomWalk}(\mathcal{V}_{\mathrm{r}}, n_{\mathrm{w}}, l_{\mathrm{w}})$, where $\mathcal{V}_{\mathrm{r}} \subseteq \mathcal{V}$ is the set of root nodes, $n_{\mathrm{w}}$ and $l_{\mathrm{w}}$ represent the number of walks per node and the walk

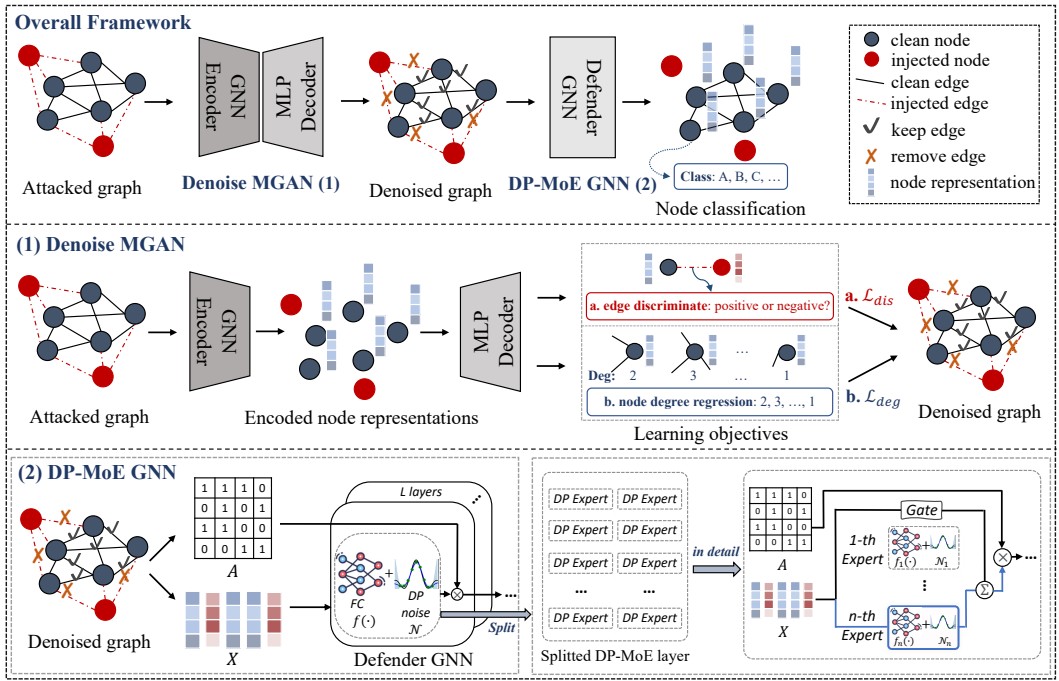

Figure 2: Our framework. First, (1) in Denoise MGAN, a cleaner graph is recovered by removing the edges connected to injected nodes, preventing their message-passing interactions with clean nodes. Second, the cleaner graph is classified using (2) in DPMoE GNN, which consists of a DP graph convolutional layer split into multiple DP expert networks with adjustable noise coefficients to handle attacks of different intensities.

lengths, respectively. By disrupting edges between a sequence of adjacent nodes, the path masking strategy encourages the model to capture structured dependencies and high-order node proximities. This enables the model to subsequently reconstruct a cleaner graph from the attacked one.

**Generate the Clean Graph via Auto-Encoder.** To construct a scalable DMGAN, we adopt GCN as the encoder and multilayer perceptrons (MLP) as the decoder. The decoder has a dual purpose: reconstructing the graph structures and predicting node degrees. To demonstrate, we first obtain the encoded representations $H$ obtained from the encoder for nodes $\mathcal{V}_v$. Then, the decoder leverages the representations of a pair of nodes as link representations to reconstruct the connections of the original graph. We use MLP to reconstruct the graph structure, and define the structure decoder as $\text{Dec}_\omega$ with weight parameters $\omega$ as follows: $\text{Dec}_\omega(h_i, h_j) = \text{Sigmoid}(\text{MLP}(h_i \circ h_j))$, where $\circ$ denotes the element-wise product, $h_i \in H$ and $h_j \in H$ are the representations of node $i$ and node $j$, respectively. Subsequently, to predict node degrees in the masked graph, we employ another MLP-based decoder for degree regression. We denote this as $\text{Dec}_\phi$ with weight parameters $\phi$: $\text{Dec}_\phi(h_v) = \text{MLP}(h_v)$, Here, $h_v$ corresponds to the representation of the target node $v \in \mathcal{V}_v$, obtained through the previous encoder. This approach allows the DMGAN module to learn the connection patterns of nodes in the original clean graph, aiding in the detection of malicious edges linked to injected nodes when the DMGAN module operates on the attacked graph.

**Self-Supervised Loss for Denoise Auto-Encoder.** The loss function for the decoder, responsible for both graph structure reconstruction and node degree regression, consists of two terms. First, a binary cross-entropy loss term is used to predict the edge for graph structure decoding:

$$\mathcal{L}_\text{e} = -\left( \frac{1}{|\mathcal{E}_\text{m}|} \sum_{(u,v) \in \mathcal{E}_\text{m}} \log(\text{Dec}_\omega(h_u, h_v)) + \frac{1}{|\mathcal{E}_\text{n}|} \sum_{(u',v') \in \mathcal{E}_\text{n}} \log(1 - \text{Dec}_\omega(h_{u'}, h_{v'})) \right), \quad (3)$$

where $h$ is the node representation output from the encoder, $\mathcal{E}_\text{n}$ is a set of nonexistent negative edges sampled from training graph $\mathcal{G}$. Second, a regression loss term is applied to measure how accurately the node degree prediction matches the original in the masked graph:

$$\mathcal{L}_\text{d} = \frac{1}{|\mathcal{V}_\text{m}|} \sum_{v \in \mathcal{V}_\text{m}} ||\text{Dec}_\phi(h_v) - \deg(v)||_2^2, \quad (4)$$

where $\deg(\cdot)$ and $\mathcal{V}_{\mathrm{m}}$ denote the masked node degree and the masked nodes in the graph, respectively. The overall objective $\mathcal{L}$ to be minimized during training is the combination of $\mathcal{L}_{\mathrm{d}}$ and $\mathcal{L}_{\mathrm{e}}$. After using the loss $\mathcal{L}$ to train the auto-encoder DMGAN, during attack evaluation, an attacked graph $\mathcal{G}' = (\mathcal{V}', \mathcal{E}', \mathcal{X}')$ is perturbed from $\mathcal{G}$ and used as an input. Then we use the trained DMGAN to decode the reconstructed edges of $\mathcal{G}'$, i.e., $\mathcal{E}_r$. We use $\mathcal{E}_r$ to index the reconstructed subgraph $\mathcal{G}_r$ from $\mathcal{G}'$, and then use $\mathcal{G}_r$ for downstream tasks.

## 4.2 Defender to Attacks of Different Intensities: DP-based Mixture of Experts

After cleaning the attacked graph with DMGAN, if any injected nodes remain in $\mathcal{G}_r$ and negatively affect predictions, we deploy the DP mechanism in the Defender GNN for robust predictions on $\mathcal{G}_r$. This involves introducing the DP mechanism into the GNN layer to create DP-GConv, which is further divided into DPMoE with different DP experts to handle various attack intensities.

**Differentially-Private Graph Convolution.** For tackling attacked graph outputs from DMGAN, we introduce Differentially-Private Graph Convolution (DP-GConv). DP-GConv updates the representation of a target node, $h_v^{(l-1)}$, output by the $(l-1)$-th layer, in three steps:

First, a Gaussian differentially-private noise module $\mathcal{M}(\cdot)$ computes Gaussian noise $\mathcal{N}$, added to $h_v^{(l-1)}$: $\mathcal{M}(h_v^{(l-1)}) = h_v^{(l-1)} + \mu\mathcal{N}$, where $\mu$ is the scaling coefficient, and $\mathcal{N}$ follows a Gaussian distribution with mean zero and standard deviation $\sigma = \sqrt{2\ln(1.25/\delta)}/\epsilon$. Second, $\mathcal{M}(h_v^{(l-1)})$ is multiplied by a learnable weight matrix $W^{(l)}$, performed by the $\text{DPLinear}(\cdot)^l$ module: $\text{DPLinear}^{(l)}(h_v^{(l-1)}) = \mathrm{W}^{(l)}\mathcal{M}(h_v^{(l-1)})$. This transformation is applied to all neighboring nodes linked with the current target node. Third, after aggregating the features of all transformed neighboring nodes, DP-GConv combines the feature of the target node and neighboring nodes, updating the target node representation from $h_v^{(l-1)}$ to $h_v^{(l)}$:

$$h_v^{(l)} = \text{COM}^{(l)} \left( \text{DPLinear}^{(l)}(h_v^{(l-1)}), \ \text{AGG} \left( \left\{ \text{DPLinear}^{(l)}(h_u^{(l-1)}), \forall u \in N_v \right\} \right) \right), \tag{5}$$

where $\text{AGG}(\cdot)$ and $\text{COM}(\cdot)$ represent neighbor aggregation and combination functions, respectively. $N_v$ is the set of all neighboring nodes $u$ of node $v$.

In our implementation, the Gaussian DP noise module $\mathcal{M}(\cdot)$ can be placed in front of DP-GConv's weight matrix, randomizing attack node features and constraining output results closer to predictions based on the original clean input, effectively neutralizing the influence of injected nodes.

**Splitted Differentially-Private Mixture-of-Experts.** To enhance the robustness of the defender GNN against node injection attacks of varying intensities, we introduce a novel approach known as Splitted Differentially-Private Mixture-of-Experts (DPMoE). This design divides each DP-GConv layer, denoted as $\text{DPLinear}(\cdot)$, into multiple expert networks. Each of these expert networks is equipped with a specific magnitude of Gaussian DP module, allowing the defender GNN to effectively handle node injection attacks of differing strengths. The DPMoE module, which is a split from $\text{DPLinear}(\cdot)$, is formulated as follows:

$$\text{DPMoE}(h) = \sum_{i \in \mathcal{T}} p_i(h) \cdot \text{DPLinear}_i(h), \tag{6}$$

where $\mathcal{T}$ represents the set of activated top-$k$ expert indices. $\text{DPMoE}(\cdot)$ combines the outputs of multiple DP expert networks, each with a different scaling coefficient $\mu_i$ for the Gaussian DP noise $\mathcal{N}$ at various magnitudes. $\mu_i$ increases linearly as the index $i$ of the expert increases. The top-$k$ activated expert indices are determined by the gate values $p_i(h)$, which can be obtained using a softmax function as described below:

$$p_i(h) = \frac{\exp(t(h)_i + \varepsilon_i)}{\sum_{k=1}^{N} \exp(t(h_i)_k + \varepsilon_k)}, \tag{7}$$

where $t(\cdot)$ denotes a linear transformation, and $N$ is the total number of experts. The activated expert indices in the DPMoE module are determined by gate values $p_i(h)$, which compute the activation probability of each expert. Logits are weighted by the $i$-th value $t(h)_i$ of the linear transformation, and a random noise term $\varepsilon_i$ is added to ensure randomness in the expert activation procedure. Typically, $\varepsilon_i$ is sampled from a Gaussian distribution.

To update the features of a target node, the DPMoE layer combines feature transformation and aggregation operations. First, it transforms the features of the target node and its neighboring nodes using $\mathrm{DPMoE}^{(l)}(\cdot)$ and then aggregates the transformed features of the neighboring nodes using $\mathrm{AGG}(\cdot)$. The final representation of the target node is obtained by combining the transformed characteristic of the target node and the aggregated characteristic of its neighboring nodes:

$$h_v^{(l)} = \mathrm{COM}^{(l)}\left(\mathrm{DPMoE}^{(l)}(h_v^{(l-1)}), \ \mathrm{AGG}\left(\left\{\mathrm{DPMoE}^{(l)}(h_u^{(l-1)}), \forall u \in N_v\right\}\right)\right). \tag{8}$$

In comparison to the previous DP-GConv formulation in Equation (5), the proposed DPMoE layer employs a gating mechanism to control multiple experts, each equipped with multi-scale DP Gaussian noises. This allows it to effectively handle node injection attacks of varying intensities while maintaining higher anti-degraded robustness.

**Theoretical Analysis of DPMoE Robustness.** Let's denote $k \in \mathcal{K}$ as the ground-truth class label for node $v$; $\epsilon, \delta$ as two differential privacy parameters of Gaussian DP mechanism; $h_v^{(l)}$ as the feature of node $v$ in layer $l$ after aggregating from the injection nodes. When Equation (9) in Proposition 1 is satisfied, we can theoretically ensure the robustness of the model.

**Proposition 1.** *Robustness Guarantee for DPMoE. For a GNN $f(\cdot)$ containing DPMoE which utilizes Gaussian DP, assume this mechanism lets the model output satisfy $(\sigma, \delta)$-DP. If the expected value $\mathbb{E}$ of the model output satisfies the following property:*

$$\mathbb{E}(f_k(h_v^{(l)})) > e^{2\epsilon} \max_{i:i \neq k} \mathbb{E}(f_i(h_v^{(l)})) + (1 + e^\epsilon)\delta, \tag{9}$$

*then the label probability output vector $p(h_v^{(l)}) = (\mathbb{E}(f_1(h_v^{(l)})), \ldots, \mathbb{E}(f_K(h_v^{(l)})))$ of $f(\cdot)$ for node $v$ satisfies the robustness: $\mathbb{E}(f_k(h_v^{(l)})) \geq \max_{i:i \neq k} \mathbb{E}(f_i(h_v^{(l)}))$.*

This proposition sheds light on how upholding DP can bolster the model's robustness against adversarial perturbations. By adhering to the $(\epsilon, \delta)$-DP constraints, the model's vulnerability to perturbations induced by adversarial injections is significantly reduced. The complete proof is in Appendix A.

In a brief proof sketch, we begin by confirming the $(\epsilon, \delta)$-DP of a GNN model that solely consists of DP-GConv. Utilizing previous propositions and equations, the proof establishes bounds on the model's expected outputs. These bounds are instrumental in deriving inequalities, which compare the expected output of the genuine class $k$ to that of any other class under adversarial perturbations. By examining these inequalities, we uncover the necessary conditions for the model to maintain robustness—specifically, that the expected output for class $k$ significantly surpasses that of any other class. The proof then extends these findings to models incorporating DPMoE, demonstrating that if a DPMoE model meets the specified conditions, its robustness is established.

## 5 EXPERIMENTS

### 5.1 SETTINGS

**Datasets.** We evaluate the robustness and scalability of our proposed DRAGON framework using the Graph Robustness Benchmark (GRB) dataset (Zheng et al., 2021), which includes graphs of varying scales, such as *grb-cora* (small-scale), *grb-citeseer* (small-scale), *grb-flickr* (medium-scale), *grb-reddit* (large-scale), and *grb-aminer* (large-scale). The statistics of the datasets are in Appendix C.

**Baselines Methods.** We evaluate the robustness of DRAGON against several baseline methods from two perspectives: First, for baselines that have been applied on robustness against injection attacks, we consider GCN-SVD (Entezari et al., 2020), GNNGuard (Zhang & Zitnik, 2020), RGCN (Zhu et al., 2019), EvenNet (Lei et al., 2022). Second, we compare against general GNN models (i.e., GCN (Kipf & Welling, 2017), GAT (Veličković et al., 2018)), RGCN (Zhu et al., 2019) and GAME (Zhang et al., 2023c) that are integrated with a generic defense approach, Adversarial Training (AT) (Madry et al., 2018). Configurations of all methods are included in Appendix C. In addition, many other graph robust learning methods such as Jaccard GCN (Wu et al., 2019b), SoftMedian (Geisler et al., 2021) and GARNET (Deng et al., 2022) are only effective for specific attacks with ad hoc designs, making the results less generalizable and practical, thus we do not include them as our primary baselines and compare them with DRAGON in Appendix G.

Table 1: Overall assessments of all methods among five GRB datasets with different-intensitiy FGSM attacks. The best result is bolded and the runner-up is underlined. Inten. denotes Attack Intensity and OOM represents Out-of-Memory. *Method*+AT means integrating the method with adversarial training.

| | Inten. | GCN | GAT | RGCN | GCNSVD | GATGuard | EvenNet | GCN+AT | GAT+AT | RGCN+AT | GAME+AT | Ours | Ours+AT |
|---|---|---|---|---|---|---|---|---|---|---|---|---|---|
| *Cora* (small) | nulla | 84.2±0.6 | 83.6±0.5 | 84.0±0.7 | 64.5±1.0 | 81.7±1.0 | 82.4±0.7 | 83.5±0.1 | 79.6±0.9 | 85.4±0.5 | **85.5±0.8** | 84.1±0.7 | 81.6±0.7 |
| | I | 65.6±3.2 | 58.9±4.2 | 51.8±1.5 | 44.1±0.4 | 80.6±1.1 | 67.7±2.2 | 82.5±1.1 | 79.4±0.8 | 79.1±2.1 | 81.7±0.4 | **84.2±0.6** | 82.0±0.7 |
| | II | 57.3±3.0 | 38.9±3.7 | 23.0±0.7 | 42.0±2.0 | 80.7±1.0 | 53.3±2.1 | 79.6±0.8 | 79.2±0.7 | 72.8±4.6 | 80.1±0.7 | **84.1±0.3** | 81.6±1.4 |
| | III | 33.6±3.0 | 24.3±4.3 | 13.6±0.3 | 37.5±2.9 | 80.7±1.0 | 39.6±1.7 | 71.4±4.7 | 79.5±0.8 | 49.7±7.2 | 79.1±0.8 | **84.0±0.5** | 81.7±0.5 |
| | IV | 17.9±0.8 | 15.8±2.5 | 12.0±0.1 | 33.7±1.4 | 79.4±1.2 | 28.9±0.8 | 47.9±3.9 | 78.3±0.1 | 25.0±2.6 | 78.3±0.2 | **84.0±0.3** | 81.0±0.9 |
| | V | 12.4±0.1 | 14.3±1.6 | 11.9±0.1 | 33.0±0.4 | 79.4±1.5 | 23.8±0.2 | 35.6±4.4 | 79.3±0.6 | 24.7±8.9 | 77.3±0.6 | **83.3±0.3** | 80.4±0.8 |
| *Citeseer* (small) | nulla | 71.6±0.8 | 72.0±0.6 | 73.7±0.7 | 68.2±1.2 | 72.8±0.3 | 69.0±0.5 | 73.8±0.6 | 69.6±1.5 | 73.9±0.3 | **76.1±1.3** | 75.9±0.6 | 71.1±0.3 |
| | I | 37.9±6.5 | 19.4±0.6 | 62.4±2.0 | 23.4±2.1 | 72.5±0.6 | 57.3±2.3 | 59.7±2.9 | 69.2±0.7 | 71.4±1.4 | 72.4±0.8 | **75.6±0.2** | 70.6±0.6 |
| | II | 33.3±8.4 | 21.1±2.5 | 45.9±5.5 | 22.6±0.6 | 72.4±0.6 | 49.0±3.0 | 28.0±1.6 | 68.8±0.7 | 64.4±1.4 | 69.3±0.9 | **74.7±0.3** | 70.4±0.7 |
| | III | 17.8±1.3 | 19.7±3.8 | 35.1±3.9 | 21.7±2.0 | 72.5±0.6 | 36.8±3.1 | 27.1±3.5 | 68.7±0.3 | 58.7±3.6 | 66.7±1.2 | **75.9±0.5** | 70.7±1.5 |
| | IV | 16.2±1.0 | 19.6±5.6 | 31.4±5.1 | 19.4±1.8 | 72.4±0.6 | 28.6±1.9 | 23.6±6.4 | 68.7±0.3 | 51.9±2.8 | 64.6±0.3 | **75.6±0.5** | 70.3±1.0 |
| | V | 21.0±4.0 | 13.5±4.7 | 32.7±2.4 | 14.6±1.0 | 72.5±0.6 | 24.0±3.3 | 26.7±9.0 | 68.8±0.8 | 47.3±4.3 | 62.8±0.7 | **75.3±0.3** | 70.6±1.1 |
| *Flickr* (medium) | nulla | 47.1±0.5 | 50.0±1.2 | 50.8±0.7 | OOM | OOM | 49.0±0.6 | 45.4±0.3 | 44.2±1.8 | 43.1±5.6 | 52.2±0.9 | **52.7±0.1** | 51.1±0.0 |
| | I | 39.6±0.9 | 47.8±2.6 | 48.2±1.0 | OOM | OOM | 48.2±0.7 | 46.3±0.8 | 44.3±2.1 | 40.3±4.6 | 45.6±1.1 | **52.0±0.2** | 51.0±0.1 |
| | II | 30.6±0.1 | 44.1±3.9 | 43.7±2.1 | OOM | OOM | 45.1±1.4 | 44.3±1.2 | 44.0±2.4 | 40.0±2.8 | 43.0±1.4 | **51.8±2.1** | 50.3±0.0 |
| | III | 13.5±1.0 | 34.2±7.3 | 18.9±2.5 | OOM | OOM | 37.2±2.3 | 27.1±7.6 | 43.1±3.1 | 45.9±3.6 | 41.1±1.1 | **51.2±1.5** | 49.8±0.0 |
| | IV | 9.6±0.3 | 24.5±9.6 | 12.6±1.3 | OOM | OOM | 29.5±3.4 | 15.2±2.7 | 42.8±3.5 | 43.5±1.2 | 40.8±0.9 | **50.3±1.8** | 48.7±0.3 |
| | V | 9.1±0.1 | 24.9±9.8 | 15.7±4.5 | OOM | OOM | 26.9±3.8 | 14.4±5.6 | 42.4±4.1 | 43.8±1.5 | 38.4±2.1 | **48.8±1.7** | 47.5±0.1 |
| *Reddit* (large) | nulla | 95.6±0.0 | 95.8±0.0 | 95.5±0.1 | OOM | OOM | 95.1±0.0 | 95.6±0.0 | 95.4±0.0 | 95.7±0.0 | 96.1±0.0 | **96.2±0.0** | 95.7±0.0 |
| | I | 95.6±0.1 | 95.4±0.1 | 93.0±0.1 | OOM | OOM | 95.0±0.1 | 95.4±0.1 | 95.4±0.2 | 93.2±0.0 | 95.3±0.0 | **96.1±0.0** | 95.6±0.1 |
| | II | 94.7±0.0 | 95.2±0.2 | 85.8±1.4 | OOM | OOM | 94.9±0.2 | 95.1±0.0 | 95.4±0.1 | 85.0±0.3 | 95.1±0.1 | **96.1±0.0** | 95.6±0.0 |
| | III | 93.6±0.1 | 94.2±0.1 | 75.4±1.4 | OOM | OOM | 93.8±0.2 | 94.4±0.2 | 95.3±0.0 | 72.0±0.1 | 94.8±0.0 | **96.0±0.0** | 95.5±0.0 |
| | IV | 91.7±0.3 | 93.9±1.0 | 59.1±1.0 | OOM | OOM | 93.7±0.1 | 93.2±0.3 | 95.2±0.1 | 51.5±0.1 | 94.2±0.2 | **95.9±0.1** | 95.5±0.0 |
| | V | 88.6±0.1 | 93.1±0.5 | 49.6±1.3 | OOM | OOM | 93.2±0.1 | 88.9±0.2 | 95.2±0.0 | 42.8±0.0 | 93.0±0.1 | **95.8±0.1** | 95.5±0.1 |
| *AMiner* (large) | nulla | 63.4±0.1 | **66.8±0.7** | 63.6±0.2 | OOM | OOM | 62.7±0.0 | 64.1±0.0 | 63.9±0.0 | 64.4±0.2 | 64.5±0.2 | 64.3±0.8 | 64.3±0.0 |
| | I | 61.0±0.5 | 59.3±0.3 | 51.2±0.6 | OOM | OOM | 61.0±0.2 | 62.7±0.0 | 63.8±0.2 | 61.0±1.0 | 63.0±0.1 | 61.7±0.5 | **64.1±0.0** |
| | II | 51.2±0.3 | 46.9±1.0 | 32.6±0.3 | OOM | OOM | 52.3±0.1 | 57.4±0.0 | 63.4±0.1 | 50.6±2.2 | 62.1±0.0 | 56.1±0.2 | **63.7±0.1** |
| | III | 39.3±0.1 | 35.6±0.8 | 21.9±0.3 | OOM | OOM | 42.7±0.0 | 48.6±0.0 | 62.7±0.0 | 38.3±1.2 | 60.8±0.1 | 49.0±0.4 | **63.1±0.1** |
| | IV | 32.2±1.2 | 30.2±2.5 | 16.51±1.2 | OOM | OOM | 32.2±0.1 | 41.3±0.0 | 62.1±0.1 | 31.0±0.7 | 58.6±0.2 | 42.5±0.3 | **62.6±0.3** |
| | V | 24.5±0.7 | 23.8±2.3 | 13.0±0.9 | OOM | OOM | 26.3±0.2 | 32.4±0.0 | 61.1±0.3 | 22.0±2.3 | 56.7±0.3 | 49.7±0.2 | **62.0±0.5** |

**Attack Strategies.** In this section, we examine five effective and distinct graph injection attack methods, including FGSM (Goodfellow et al., 2015), SPEIT (Zheng et al., 2020), PGD (Madry et al., 2018), TDGIA (Zou et al., 2021), HAO (Chen et al., 2022), that can degrade the performance of victim GNNs, including our proposed DRAGON framework and other baselines. Note that we focus primarily on injection attacks, which have received considerable attention due to their ease of deployment and cost-effectiveness compared to modifying the original graph input.

In configuring attack strategies, we maintain consistency with the default GRB configuration, varying attack intensities by multiplying the number of injected nodes to create five intensities from 1 to 5 as detailed in Appendix C.1, while keeping hyperparameters consistent with GRB for generating attacks and surrogate models, with additional specifics in Appendix C.2. The code can be accessed through `https://github.com/chunhuizng/emergent-degradation`.

## 5.2 OVERALL ROBUSTNESS AND SCALABILITY ON GRB

We first assess DRAGON's robustness and scalability, comparing it against state-of-the-art (SOTA) baseline methods. Results are summarized in Table 1. Table 1 reveals DRAGON's comprehensive superiority over other baselines in terms of robust accuracy across five distinct attack intensities (ranging from $I$ to $V$). Additionally, DRAGON achieves these results without encountering out-of-memory issues while handling five graph datasets of varying scales on the same hardware platform.

Specifically, with regards to anti-degraded robustness, DRAGON demonstrates consistent and impressive robust accuracy under five different attack intensities among various datasets, while other baseline methods, including robust GNNs, adversarial trained GNNs, and general GNNs, experience severe degradation in robustness as the attack intensity increases. For instance, on the small-scale grb-citeseer dataset, DRAGON outperforms the most competitive baseline GAME+AT by 3.2% when the attack intensity is $I$ and by 12.5% when the attack intensity increases to $V$; On the medium grb-flickr dataset, compared to the most competitive baseline GAME+AT, DRAGON outperforms this baseline by 6.4% when the attack intensity is $I$ and by 10.4% when the attack intensity increases to $V$; On the large grb-reddit dataset, DRAGON outperforms the most competitive baseline, GAT+AT, by 0.7% when the attack intensity is $I$ and by 0.6% when the attack intensity increases to $V$; In addition,

our DRAGON also demonstrates strong representation ability on clean graphs and outperforms other baselines in the nulla (i.e., without attack) setting. Moreover, some baseline methods, such as GCNSVD and GATGuard, struggle with GPU memory limitations due to their reliance on heavy computation complexity and run into out-of-memory problems on medium and large datasets, unlike our DRAGON, which shows reliable scalability. We note that DRAGON exhibits relatively lower robustness on the AMiner dataset and we provide the explanation in Appendix B.

## 5.3 ABLATION STUDY

We examine the contributions of DRAGON's individual components, namely the denoising preprocessing component (DMGAN) and the defender component (DPMoE), we perform ablation experiments. We systematically remove each component to observe its impact on DRAGON's performance. The ablated models are denoted as: (a) without denoise, (b) without DPMoE, and (c) the GNN backbone, representing the base model without both denoise and DPMoE, as shown in Table 2 (a fine-grained ablation study on DMGAN is in Appendix D).

Our experiments reveal that removing any component from DRAGON leads to decreased performance in node injection attacks, particularly as the attack intensity increases. This emphasizes the critical role each component plays in enhancing the model's robustness and providing anti-degraded robustness:

For (a) w.o. denoise, eliminating the DMGAN component significantly impairs DRAGON's ability to recover clean graph information from

Table 2: Ablation studies for DRAGON on graphs of varying scales and under FGSM attack of varying intensities. Base model denotes backbone w.o. DRAGON.

|  | Int. | Base model | w.o. denoise | w.o. DPMoE | DRAGON |
|---|---|---|---|---|---|
| *Cora* *(small)* | nulla | 83.1±0.5 | **84.2±0.2** | 83.9±0.6 | 84.1±0.3 |
|  | I | 81.6±4.2 | 84.1±1.3 | 81.8±1.4 | **84.2±0.4** |
|  | II | 80.7±3.7 | 83.9±1.9 | 80.9±0.8 | **84.1±1.6** |
|  | III | 79.7±4.3 | 83.9±3.7 | 80.1±0.7 | **84.0±2.0** |
|  | IV | 76.4±2.5 | 83.5±1.3 | 77.0±1.6 | **84.0±1.3** |
|  | V | 75.2±1.6 | 83.0±2.1 | 76.1±2.3 | **83.3±2.9** |
| *Citeseer* *(small)* | nulla | 74.1±0.6 | **76.0±0.5** | 75.8±0.3 | 75.9±0.5 |
|  | I | 73.4±0.6 | 75.5±3.1 | 73.5±1.1 | **75.6±1.8** |
|  | II | 72.7±2.5 | 75.6±0.7 | 72.9±2.0 | **74.7±2.1** |
|  | III | 71.3±3.8 | 75.7±3.5 | 71.7±1.2 | **75.9±1.8** |
|  | IV | 70.5±5.6 | 75.4±2.5 | 70.8±2.7 | **75.6±0.7** |
|  | V | 69.6±4.7 | 75.0±2.4 | 71.1±3.2 | **75.3±2.4** |
| *AMiner* *(large)* | nulla | **66.8±0.1** | 66.3±0.1 | 64.8±0.0 | 64.3±0.8 |
|  | I | 59.3±0.5 | 61.7±0.4 | 60.9±0.8 | **61.7±0.5** |
|  | II | 46.9±0.3 | 52.4±0.7 | 51.9±0.2 | **56.1±0.2** |
|  | III | 35.6±0.1 | 42.4±1.7 | 40.9±0.4 | **49.0±0.4** |
|  | IV | 30.1±1.2 | 34.0±0.5 | 34.6±0.3 | **42.5±0.3** |
|  | V | 23.8±0.7 | 26.5±2.2 | 28.1±0.9 | **39.7±0.2** |

an attacked graph input. For instance, on the small grb-cora dataset, the model experiences a 0.1% loss in accuracy at an attack intensity of $I$ and a 0.3% loss at an intensity of $V$. This illustrates that a cleaner graph reconstructed by DMGAN is easier for defense. For (b) w.o. DPMoE, disabling the DPMoE component entirely hinders DRAGON's capacity to manipulate Gaussian DP noises of varying magnitudes to counteract attacks of different intensities. On the grb-citeseer, the removal of this component results in a 2.1% loss in accuracy at an attack intensity of $I$ and a 4.2% loss at an intensity of $V$. This demonstrates that DPMoE enhances the robustness of GNNs against adversarial node injections of varying strengths by providing diverse yet effective Gaussian DP noises. Finally, for (c) w.o. denoise and DPMoE, removing both components reduces DRAGON to a vanilla GAT model. On the larger grb-aminer, this leads to a 2.5% increase in accuracy at an attack intensity of $I$ and a 15.9% loss at an intensity of $V$.

Above results highlight the effectiveness of DRAGON in denoising and enhancing a base model's capacity to learn robust representations against attacks of different intensities.

## 5.4 WHAT WINS ANTI-DEGRADED ROBUSTNESS?

We investigate the denoising module (DGMGAN) in recovering attacked graphs and the capability of the defender component (DPMoE) in manipulating the magnitude of the Gaussian Differential Privacy mechanisms and analyze how they improve DRAGON's anti-degradation robustness.

**Denoising Injected Nodes by DMGAN.** To gauge the effectiveness of DMGAN in DRAGON, we visually compare the attacked graph input with the denoised version produced by DMGAN in Figure 3, and specific statistics for this edge denoising process are listed in Appendix D. The results illustrate that as the attack intensity increases, DMGAN continues to efficiently remove numerous malicious edges associated with injected nodes. This reduction in the negative impact of the attack showcases DMGAN's robustness against node injection attacks of varying intensities.

Before delving into the DPMoE design, it is crucial to acknowledge that relying solely on the DP mechanism within a standard GNN backbone, without MoE, poses challenges in striking a balance between performance in attack and non-attack scenarios. As shown in Figure 4, models with higher DP scaling coefficients display flatter performance curves, indicating robustness in high-intensity attack scenarios but reduced accuracy in non-attack scenarios.

**Effectiveness of DPMoE.** To attain a more favorable trade-off between non-attack and attack scenarios of varying intensity, DPMoE is introduced to dynamically balance these scenarios. This is achieved by employing multiple experts that integrate DP scaling coefficients of varying magnitudes. Such an approach allows the activation of the appropriate expert network with the corresponding DP scaling coefficient, yielding an (more) optimal solution for both non-attack and varying-intensity attack scenarios: (i) in non-attack scenarios, DPMoE activates the expert with minimal DP noise; (ii) as attack intensity increases, it activates the expert with larger DP noise magnitudes. This adaptive adjustment of the DP scaling coefficient by DPMoE for non-attack and varying-intensity attack inputs results in impressive performance under varying-intensity attack evaluations and non-attack evaluations.

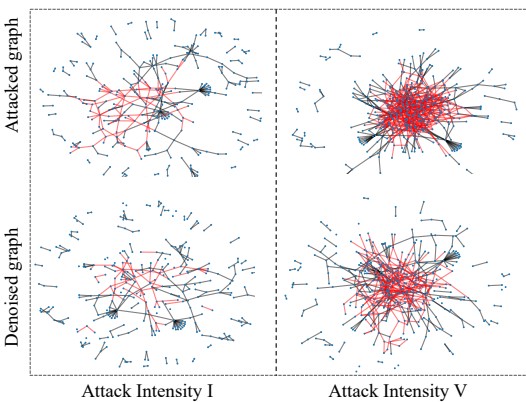

Figure 3: The visualization of sampled *Cora* graph at attack intensity I and V and their DMGAN-denoised version. The black edges denote normal edges and the red edges represent malicious edges.

In addition to the above experiments, we present more comprehensive findings and clarifications, including evaluations against SPEIT, PGD, TDGIA, and HAO attacks, in Appendix B. Appendix C elucidates all dataset and attack strategy settings. Further insight into the trade-offs within the denoise module is offered in Appendix D. Additionally, Appendix E and F delve into discussions about GNNs under modification and adaptive attacks, although these discussions slightly extend beyond the scope of this paper, as explained in the appendix. Detailed analyses demonstrating our method's linear time and memory complexities, contributing to its scalability and efficiency, can be found in Appendix H. Furthermore, the realistic importance of considering high-intensity graph attacks in the real world is discussed in Appendix I.

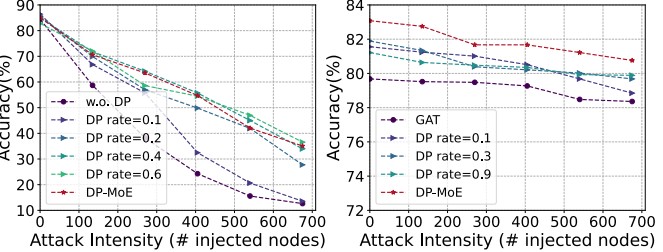

Figure 4: Different DP rates (scaling coefficients) on DRAGON w. single DP rate and w. multiple DP rates via DPMoE using standard training (left) and adversarial training (right) on *Cora* dataset.

## 6 CONCLUSION

We begin by identifying two practical issues: severe robustness degradation and limited scalability in current adversarial graph learning. Then to address them simultaneously, we propose a novel framework named DRAGON by utilizing a denoising masked graph auto-encoder and a differential privacy mechanism. Our experimental results show the effectiveness of DRAGON in denoising malicious edges, counteracting the injected attack node features through differential privacy, and navigating graph data of varying scales and attacks of different intensities. Overall, DRAGON is a robust and scalable solution for avoiding GNN's emergent robustness degradation in unpredictable real-world applications.

ETHICS STATEMENTS

We emphasize the critical importance of considering high-intensity graph attacks and the emergent severe robustness degradation, which is a bitter lesson from unpredictable real-world scenarios. We discuss this in detail in Appendix I. Therefore, we improve the robustness of GNNs with respect to severe robustness degradation, ensuring their effectiveness against adversarial attacks even in high-intensity scenarios and with large graph datasets.

Furthermore, advances in graph adversarial learning may lead to potential vulnerabilities in GNNs. Therefore, it is imperative to develop robust graph defense methods and advocate for responsible GNN deployment and management. Our novel defense framework, DRAGON, paves the way for future research on trustworthy graph learning. We emphasize the need for comprehensive strategies to address possible implications and threats in future research and applications.

ACKNOWLEDGMENTS

This work was partially supported by Dartmouth, the NSF under grants IIS-2321504, IIS-2334193, IIS-2203262, IIS-2217239, CNS-2203261, IIS-2340346, and CMMI-2146076. Any opinions, findings, conclusions, or recommendations expressed in this material are those of the authors and do not necessarily reflect the views of the sponsors.

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

## A    PROOF

This section provides a rigorous proof and analysis to elucidate the robustness of the DPMoE module. The proof of Lemma 1 is adapted from Lemma 1 in the research (Lecuyer et al., 2019), aligning the notations and the scope of this paper.

We first assume the composition model $f(\cdot)$ with classical Gaussian mechanism satisfies $(\epsilon, \delta)$-DP property. This assumption follows precedents in NLP (Wang et al., 2021) and CV (Lecuyer et al., 2019), where language/vision models composed of the DP module can be assumed to satisfy $(\epsilon, \delta)$-DP. The $(\epsilon, \delta)$-DP can be justified through its use of the classical Gaussian DP output perturbation (this mechanism is inherently $(\epsilon, \delta)$-DP), due to the post-processing invariance of differential privacy (i.e., any computation on the output of the DP mechanism remains DP).

Given $B_p(r)$ representing a $p$-norm ball with radius $r$, defined as:

$$B_p(r) = \{\Delta h_v^{(l)} \in \mathbb{R}^N : \|\Delta h_v^{(l)}\|_p \leq r\}, \tag{10}$$

where $\Delta h_v^{(l)}$ represents the noise affecting $h_v^{(l)}$ at layer $l$ after message passing for node $v$. Suppose a node classification model $f(\cdot)$ is robust to this perturbation if, for all $\Delta h_v^{(l)} \in B_p(r)$,

$$f_k(h'^{(l)}_v) > \max_{i:i \neq k} f_i(h'^{(l)}_v). \tag{11}$$

**Lemma 1.** *(Expected Output Bound)*
*Let $f(\cdot)$ be a GNN incorporating DP-GConv and having bounded output $f(h_v^{(l)}) \in [0, b]$, $b \in \mathbb{R}^+$, that satisfies $(\epsilon, \delta)$-DP. Then, the expected output $\mathbb{E}[f(h_v^{(l)})]$ has the following property:*

$$\mathbb{E}[f(h_v^{(l)})] \leq e^\epsilon \mathbb{E}[f(h'^{(l)}_v)] + b\delta, \quad \forall \Delta h_v^{(l)} \in B_p(b). \tag{12}$$

*Proof.* For $\Delta h_v^{(l)} \in B_p(b)$ that modifies $h_v^{(l)}$ to $h'^{(l)}_v$, the expected output of $f(\cdot)$ is

$$\mathbb{E}[f(h_v^{(l)})] = \int_0^b Pr(f(h_v^{(l)}) > t)dt. \tag{13}$$

Using the post-processing property of $(\epsilon, \delta)$-DP algorithms and applying this to $f(\cdot)$, we deduce:

$$\mathbb{E}[f(h_v^{(l)})] \leq e^\epsilon \left( \int_0^b Pr(f(h'^{(l)}_v) > t)dt \right) + b\delta \tag{14}$$

$$= e^\epsilon \mathbb{E}[f(h'^{(l)}_v)] + b\delta.$$

$\square$

Assuming $f(\cdot)$ incorporating DP-GConv satisfies $(\epsilon, \delta)$-DP and $f(h_v^{(l)}) = (f_1(h_v^{(l)}), \ldots, f_K(h_v^{(l)}))$, where $f_k(h_v^{(l)}) \in [0, 1]$, applying Lemma 1 with $b = 1$, we have

$$\mathbb{E}[f_k(h_v^{(l)})] \leq e^\epsilon \mathbb{E}[f_k(h'^{(l)}_v)] + \delta, \quad \forall k, \forall \Delta h_v^{(l)} \in B_p(1). \tag{15}$$

**Proposition 1.** *Robustness Guarantee for DPMoE. For a GNN $f(\cdot)$ containing DPMoE which utilizes Gaussian DP, assume this mechanism lets the model output satisfy $(\sigma, \delta)$-DP. If the expected value $\mathbb{E}$ of the model output satisfies the following property:*

$$\mathbb{E}(f_k(h_v^{(l)})) > e^{2\epsilon} \max_{i:i \neq k} \mathbb{E}(f_i(h_v^{(l)})) + (1 + e^\epsilon)\delta, \tag{16}$$

*then the label probability output vector $p(h_v^{(l)}) = (\mathbb{E}(f_1(h_v^{(l)})), \ldots, \mathbb{E}(f_K(h_v^{(l)})))$ of $f(\cdot)$ for node $v$ satisfies the robustness: $\mathbb{E}(f_k(h_v^{(l)})) \geq \max_{i:i \neq k} \mathbb{E}(f_i(h_v^{(l)}))$.*

*Proof.* We first consider a GNN, $f(\cdot)$, containing only DP-GConv. By Equation (15), derived from Lemma 1, we have:

$$\mathbb{E}(f_k(h_v^{(l)})) \leq e^\epsilon \mathbb{E}(f_k(h'^{(l)}_v)) + \delta, \tag{17}$$

$$\mathbb{E}(f_i(h'^{(l)}_v)) \leq e^\epsilon \mathbb{E}(f_i(h_v^{(l)})) + \delta. \tag{18}$$

Given the lower-bound on $\mathbb{E}(f_k(h'^{(l)}_v))$ by Equation (17) and the upper-bound on $\max_{i \neq k} \mathbb{E}(f_i(h'^{(l)}_v))$ by Equation (18), we deduce:

$$
\begin{aligned}
\mathbb{E}(f_k(h'^{(l)}_v)) \overset{equation\ 17}{\geq} & \frac{\mathbb{E}(f_k(h^{(l)}_v)) - \delta}{e^\epsilon} \\
> & \frac{e^{2\epsilon} \max_{i:i \neq k} \mathbb{E}(f_i(h^{(l)}_v)) + (1 + e^\epsilon)\delta - \delta}{e^\epsilon} \\
= & e^\epsilon \max_{i:i \neq k} \mathbb{E}(f_i(h^{(l)}_v)) + \delta \\
\geq & \max_{i:i \neq k} \mathbb{E}(f_i(h'^{(l)}_v)),
\end{aligned}
\tag{19}
$$

aligning with Equation (20), implying robustness of the model $f(\cdot)$ containing DP-GConv.

To further prove the robustness of $f(\cdot)$ containing DPMoE, we simplify Equation (8) to $h^{(l+1)}_v =$ Update$(h^{(l)}_v)$ and substitute it into Equation (16), to yield:

$$
\mathbb{E}[\text{Update}(f_k(h^{(l)}_v))] > e^{2\epsilon} \max_{i:i \neq k} \mathbb{E}[\text{Update}(f_i(h^{(l)}_v)) + (1 + e^\epsilon)\delta. \tag{20}
$$

Thus, when $f(\cdot)$ containing DPMoE satisfies Equation (20), it satisfies robustness. Note that we set $\mu_i$ to 1 to obtain a more concise proof. $\qquad\square$

Proposition 1 explicates the interrelation between the model's robustness and the perturbations in the features. This interrelation enables the identification of a maximum solution, denoted as $\Delta h^{(l)}_{v(\max)}$, ensuring the model's robustness. Post injection attacks, let's denote $h^{(l)}_v$ as:

$$
h^{(l)}_v = \text{COM}^{(l)} \left( \text{DPMoE}^{(l)} h^{(l-1)}_v, \text{AGG} \left( \left\{ \text{DPMoE}^{(l)} h^{(l-1)}_u, \text{DPMoE}^{(l)} h^{(l-1)}_w \right\} \right) \right), \tag{21}
$$

where $\forall u \in N_v \wedge u \in G$, and $\forall w \in N_v \wedge w \in G' \wedge w \notin G$. Denote the attack method as $\text{Att}(\cdot)$ and the altered graph as:

$$
G' = \text{Att}(\Delta E, \Delta N, G), \tag{22}
$$

where $\Delta E$ and $\Delta N$ represent the budget of edges per injected node and the budget of injected nodes, respectively, and $G$ is the input graph.

According to Proposition 1, there exists a maximum solution $\Delta h^{(l)}_{v(\max)}$ that certifies the robustness of the model, provided Equation (16) holds true, and the $(\epsilon, \delta)$-DP in DP-GConv is maintained. Here, $\Delta h^{(l)}_{v(\max)}$ = Equation (8) - Equation (21). By solving Equations (16) and (21) along with $\Delta h^{(l)}_{v(\max)}$, a maximum $\Delta N$ can also be found to guarantee model robustness with a fixed $\Delta E$. Importantly, there is no necessity to inversely solve for the explicit maximum $\Delta N$.

## B  ADDITIONAL PERFORMANCE COMPARISONS

We also test the robustness of DRAGON under the PGD attack and the SPEIT attack. The results are shown in Figure 6, 7 and 5. The figures show our method outperforms all baselines under representative attacks as a scalable and robust framework for GNNs.

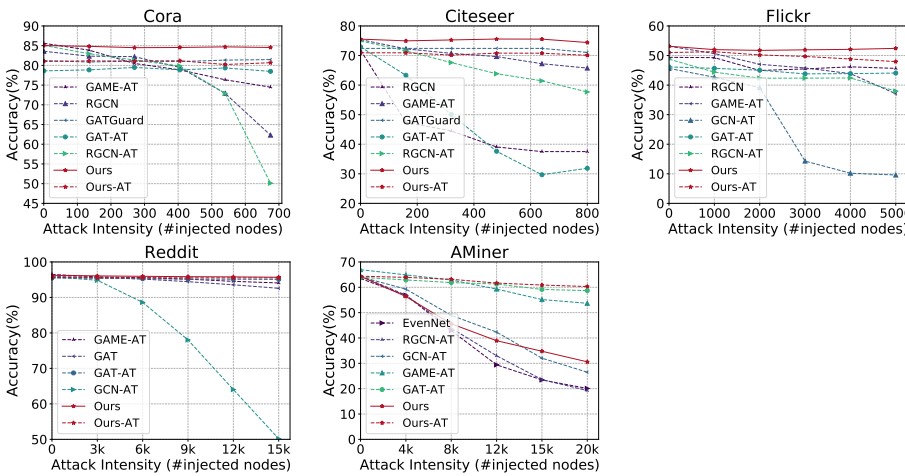

Figure 5: The performance of top-5 baselines and our method under the SPEIT Attack.

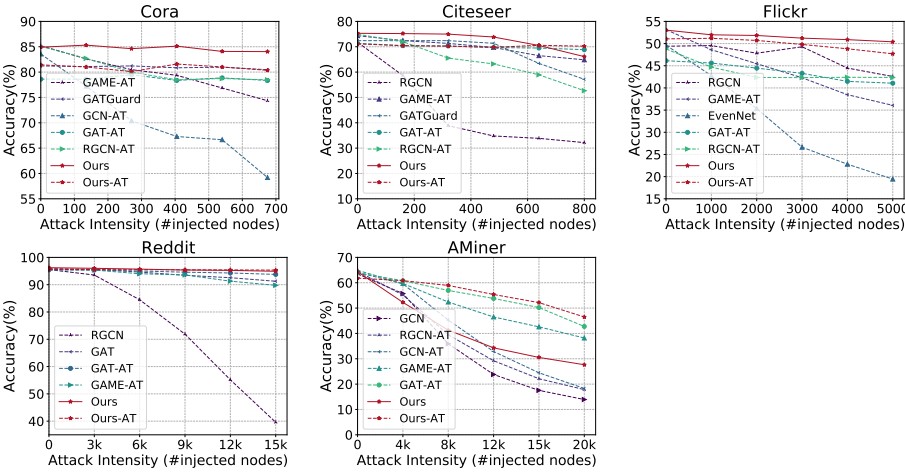

Figure 6: The performance of top-5 baselines and our method under the PGD Attack.

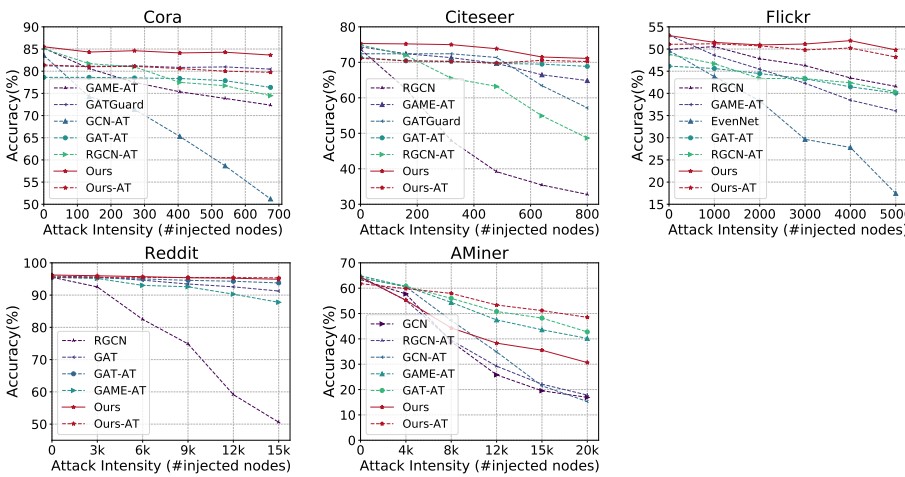

Figure 7: The performance of top-5 baselines and our method under the TDGIA Attack.

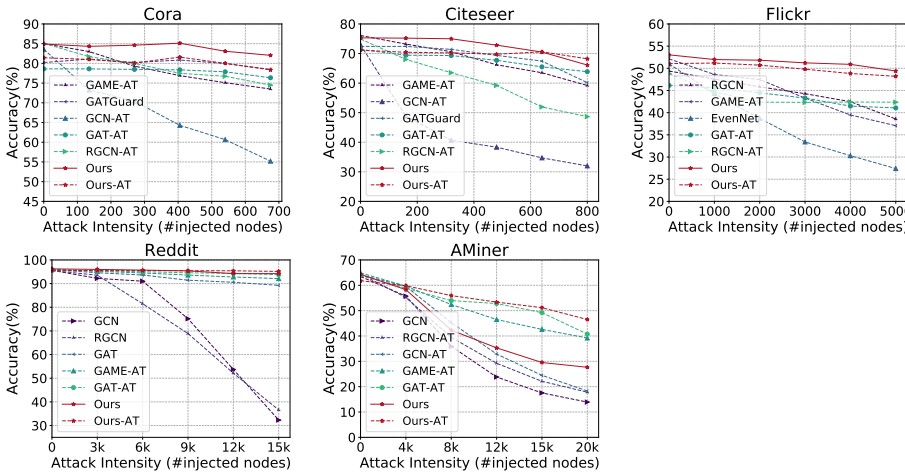

Figure 8: The performance of top-5 baselines and our method under the HAO Attack.

We note that DRAGON exhibits relatively lower robustness on the AMiner dataset compared to others. This outcome can be attributed to the distinctive characteristics of our DPMoE module. By incorporating DP noise into the feature space, our approach effectively mitigates the adverse effects stemming from the malicious aggregation of injected node features. This mechanism enhances robustness, especially on datasets with larger feature dimensions, as evidenced by our empirical findings. Notably, the AMiner dataset has a modest feature dimension (#Feat 100). In contrast, datasets like Citeseer (#Feat 768) and Reddit (#Feat 602), characterized by richer feature dimensions, show minimal performance degradation. This highlights the strength of our method in handling datasets with more extensive feature information - a common scenario in real-world data.

# C    IMPLEMENTATION DETAILS

## C.1    REPRODUCIBILITY SETTINGS

**Training and Evaluation Configuration.** In our experiments, we maintain consistency with the default hyperparameters of the GRB benchmark for the baseline methods. To ensure reliable results, we conduct 5 runs for each experimental result and report the mean value and standard deviation. Additionally, we adhere to the GRB benchmark's data splitting protocol, with 60% of the graph data as the training set, 10% as the validation set, and 30% as the test set for each benchmark dataset. Statistics of GRB data covering small to large graphs are listed in Table 4. All experiments are performed on an NVIDIA V100 GPU with 32 GB of memory.

To ensure reproducibility, we follow the hyperparameter settings of baselines used in GRB (Zheng et al., 2021) and other original papers (Chen et al., 2022; Zhang et al., 2023c). The hyperparameters of DPMoE are given in Table 5, including hyperparameters of adversarial training listed in Table 3 Specifically, for *Cora*, we set the total number of experts to $N = 10$ and the number of activated experts to $k = 2$. For other datasets, by default, we set the total number of experts to $N = 4$ and the number of activated experts to $k = 1$. Note that the DP scaling coefficient $\mu_i$ for each individual expert is *linearly* increased via multiplying the minimum coefficient by $i$ as the index $i$ of that expert increases. The number of experts is determined by evaluating the performance of the validation set. The experiment results on the Cora dataset in Table 6 assess the DPMoE module's performance with varying numbers of experts. We find that when the number of experts (num) is set to 10, the accuracy of the validation set approaches a marginal effect. However, when the number of experts exceeds 10, uncertainty increases, potentially leading to instability in optimizing a larger number of expert models. Therefore, we set the number of expert models to 10, which allows us to achieve acceptable differential privacy rates.

Additionally, hyperparameters of DMGAN are shown in Table 10. The mask rate is 0.7, the walks per node is 1, and the walk length is 3.

Table 3: Hyperparameters for adversarial training. The settings follow the GRB benchmark.

| Datasets | Cora | Citeseer | Flickr | Reddit | AMiner |
|---|---|---|---|---|---|
| Injections | 20 | 30 | 200 | 500 | 500 |
| Edges | 20 | 20 | 100 | 200 | 100 |
| Step size | 0.01 | 0.01 | 0.01 | 0.01 | 0.01 |
| Iteration | 10 | 10 | 10 | 10 | 10 |
| Attack | FGSM | FGSM | FGSM | FGSM | FGSM |

Table 4: Statistics of GRB data covering small to large graphs.

| GRB Datasets | Scale | Nodes | Edges | Feat. | Classes | Feat. Range (normalized) |
|---|---|---|---|---|---|---|
| Cora | Small | 2,680 | 5,148 | 302 | 7 | [-0.94, 0.94] |
| Citeseer | Small | 3,191 | 4,172 | 768 | 6 | [-0.96, 0.89] |
| Flickr | Medium | 89,250 | 449,878 | 500 | 7 | [-0.47, 1.00] |
| Reddit | Large | 232,965 | 11,606,919 | 602 | 41 | [-0.98, 0.99] |
| AMiner | Large | 659,574 | 2,878,577 | 100 | 18 | [-0.93, 0.93] |

Table 5: Hyperparameters of the DPMoE defender GNN.

| Datasets | Layer num. | Hidden size | Heads | Dropout | LR | | Backbone layer | | Minimal DP rate | |
|---|---|---|---|---|---|---|---|---|---|---|
| | | | | | w.o. AT | w. AT | w.o. AT | w. AT | w.o. AT | w. AT |
| Cora | 3 | 128 | 4 | 0.5 | 0.001 | 0.01 | GATGuard | GAT | 0.3 | 0.1 |
| Citeseer | 3 | 64 | 6 | 0.5 | 0.001 | 0.01 | GATGuard | GAT | 0.3 | 0.1 |
| Flickr | 3 | 64 | 8 | 0.5 | 0.0001 | 0.0001 | GAT | GAT | 0.3 | 0.1 |
| Reddit | 2 | 64 | 8 | 0.5 | 0.01 | 0.01 | GAT | GAT | 0.1 | 0.1 |
| AMiner | 3 | 64 | 4 | 0.5 | 0.01 | 0.01 | GAT | GAT | 0.1 | 0.1 |

Table 6: The results of DPMoE module with different numbers of experts on Cora dataset under FGSM attack I.

| num. of experts | 3 | 5 | 10 | 75 | 150 |
|---|---|---|---|---|---|
| Accuracy | 0.825 | 0.841 | 0.842 | 0.841 | 0.839 |

## C.2 MORE DETAILS ABOUT ATTACK STRATEGIES

We examine three effective and diverse graph attack methods that can impair the performance of victim GNNs, including our proposed DRAGON framework and other baselines. The details of these attack strategies are listed as follows:

- **FGSM**: The Fast Gradient Sign Method (FGSM) (Goodfellow et al., 2015) computes the optimal max-norm constrained perturbation as attacks by linearizing the loss function around the current value of the parameters.

- **SPEIT**: SPEIT (Zheng et al., 2020) emerges as the first-place winner in the KDD-CUP 2020 Graph Adversarial Attack & Defense competition. It generates perturbed adjacent matrix and feature gradient attacks for a global black-box node injection attack.

- **PGD**: The Projected Gradient Descent (PGD) (Madry et al., 2018) is an adversary method that leverages local first-order gradient information about the network to generate the strongest attack inputs.

- **TDGIA**: Topological Defective Graph Injection Attack (Zou et al., 2021) introduces the topological defective edge selection strategy and the smooth feature optimization objective to generate the features for the injected nodes.

- **HAO**: Harmonious Adversarial Objective (HAO) (Chen et al., 2022) introduces homophily unnoticeability that enforces graph injection attack to preserve the homophily, thereby enabling stronger attacks.

In order to generate gradient-based attacks, we follow previous research (Zheng et al., 2021) and use GCN as the surrogate model. To enhance the strength, stability, and transferability of the attacks, we also add layer normalization (LN) layers provided by the GRB benchmark to the GCN surrogate model. The hyperparameters of the surrogate model are presented in Table 7. The step size is 0.01 and iteration is 1000 when attacks are generated. In addition, We provide details of node injection attack intensities from $nulla$ to V in Table 8.

Regarding attacks, in our setting, integrating other single-node injection attacks such as SNI (Tao et al., 2021) as a baseline proves challenging due to limited edge scope, which differs significantly from other baselines that involve injected nodes with multiple edges. Furthermore, Table 9 shows that these baselines exhibit stronger attacks than Tao et al. (2021)'s method. Notably, our method's robustness is demonstrated against the latest and most advanced injection attacks in Appendix B.

Table 7: Hyperparameters of the surrogate model (the settings follow the GRB benchmark).

| Datasets | Cora | Citeseer | Flickr | Reddit | AMiner |
|---|---|---|---|---|---|
| Hidden Size | 64 | 64 | 128 | 128 | 128 |
| Layer Number | 3 | 3 | 3 | 3 | 3 |
| Learning Rate | 0.01 | 0.01 | 0.01 | 0.01 | 0.01 |
| Dropout | 0.5 | 0.5 | 0.5 | 0.5 | 0.5 |

Table 8: The configurations about attacks of five intensities and without attacks: # node represents the number of injected nodes, # edge/n. represent the max number of edges per injected node, and nulla represents *without* any attack injections into the graph input.

| Dataset | Injection | Attack Intensity | | | | | |
|---|---|---|---|---|---|---|---|
| | | nulla | I | II | III | IV | V |
| Cora | # node | 0 | 1*135 | 2*135 | 3*135 | 4*135 | 5*135 |
| | # edge/n. | 0 | 20 | 20 | 20 | 20 | 20 |
| Citeseer | # node | 0 | 1*160 | 1*160 | 1*160 | 1*160 | 1*160 |
| | # edge/n. | 0 | 20 | 20 | 20 | 20 | 20 |
| Flickr | # node | 0 | 1*1000 | 2*1000 | 3*1000 | 4*1000 | 5*1000 |
| | # edge/n. | 0 | 100 | 100 | 100 | 100 | 100 |
| Reddit | # node | 0 | 1*3000 | 2*3000 | 3*3000 | 4*3000 | 5*3000 |
| | # edge/n. | 0 | 200 | 200 | 200 | 200 | 200 |
| AMiner | # node | 0 | 1*4000 | 2*4000 | 3*4000 | 4*4000 | 5*4000 |
| | # edge/n. | 0 | 100 | 100 | 100 | 100 | 100 |

Table 9: DRAGON's performance under attacks at Intensity V. Other methods exhibit stronger attacks than Single Node Injection Attack (SNI) (Tao et al., 2021).

| | SNI | FGSM | SPEIT | PGD | TDGIA | HAO |
|---|---|---|---|---|---|---|
| Ours | 52.5 | 48.8 | 51.2 | 51.3 | 48.7 | **48.1** |

## D  TRADE-OFFS OF DMGAN

According to Figure 9, the trade-off associated with using DMGAN as a denoising module needs to be considered: there is a risk of removing the original edges along with the malicious edges, which may affect the performance of the model when it is used in a non-attack setting. After careful study, we find that a reconstruction rate of 0.3 provides a good balance between these trade-offs. For example, compared with DRAGON integrated with a vanilla graph autoencoder, our DRAGON integrated with DMGAN sacrifices only little accuracy under non-attack settings, but shows a larger accuracy improvement when the graph is under attack. The improvement of our DRAGON increases as the attack intensity increases, demonstrating the robustness of our DPMoE design from several perspectives.

To further explore the denoise ability of DMGAN, we conduct the experiment and report the results in Table 11. Let's denote TPR-N↑ (FNR-N↑) as TPR (FNR) for link prediction between normal/original

nodes; and TPR-A↑ (FNR-A↑) as TPR (FNR) for link prediction between original nodes and injection/attack nodes. FNR-N (less than 7.4%) indicates how many edges between original nodes are wrongly removed by DMGAN. According to the previous research (Jin et al., 2018), the edge noise of this level between original nodes only causes a small robustness degradation (less than 3%) on GNNs. FNR-A (more than 25.1%) indicates how many more harmful malicious edges of attack nodes are removed by DMGAN.

In order to explore the sensitivity of hyperparameters of DMGAN and how each loss term contributes to denoising on the attacked graph, we present sensitivity analysis in Table 12, Figure 10 and a fine-grained ablation study in Table 13.

Table 10: Hyperparameters of DMGAN. We use a simple graph auto-encoder model design to ensure scalability.

| Auto-encoder | Model | Layers Num. | Hidden Size | LR | Dropout |
|---|---|---|---|---|---|
| Encoder | GCN | 1 | 128 | 0.01 | 0.8 |
| Decoder | MLP | 2 | 64 | 0.01 | 0.2 |

Table 11: TPRs and FNRs of DMGAN under FGSM attack on Cora dataset.

| | nulla | I | II | III | IV | V |
|---|---|---|---|---|---|---|
| TPR-N↑ | 0.954 | 0.941 | 0.935 | 0.931 | 0.929 | 0.926 |
| FNR-N↓ | 0.046 | 0.059 | 0.065 | 0.069 | 0.071 | 0.074 |
| TPR-A↓ | N.A. | 0.638 | 0.674 | 0.691 | 0.724 | 0.749 |
| FNR-A↑ | N.A. | 0.362 | 0.326 | 0.309 | 0.276 | 0.251 |

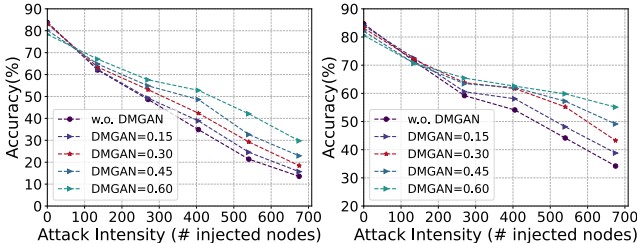

Figure 9: The effect of DMGAN with different reconstruction rates on the performance of GAT (left) and DPMoE (right) as defender GNNs in the Cora dataset under the FGSM attack (the DPMoE module uses the GAT layer as the backbone layer of each expert network).

Table 12: Sensitivity analysis for hyperparameter of DMGAN loss $\alpha$ of $L = L_d + \alpha L_e$.

| $\alpha$ | 0.000 | 0.005 | 0.010 | 0.015 | 0.020 |
|---|---|---|---|---|---|
| DMGAN | 48.31 | **48.63** | 48.56 | 48.48 | 48.41 |
| DRAGON | 51.63 | **52.02** | 51.90 | 51.88 | 51.87 |

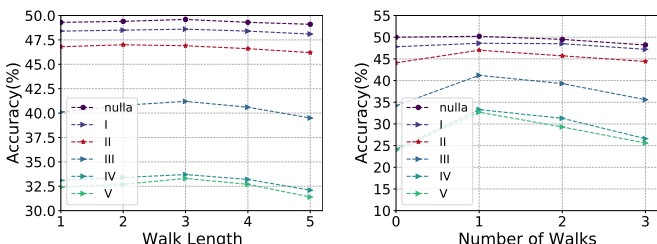

Figure 10: The effect of DMGAN with different walk lengths and number of walks under FGSM attack on Flickr datasets. I-V represents attack intensities.

Table 13: Fine-grained Ablation studies under FGSM attack of varying intensities on Flickr dataset. DM-GAN(Link) denotes only using link prediction loss in DMGAN and DMGAN(Link + Degree) denotes incorporating both link prediction loss and degree regression loss.

| Models | nulla | I | II | III | IV | V |
|---|---|---|---|---|---|---|
| Vanilla GAT | 50.0 | 47.8 | 44.1 | 34.2 | 24.5 | 24.9 |
| +DPMoE | 52.9 | 51.6 | 49.4 | 49.1 | 48.5 | 47.7 |
| +DMGAN(Link) | 49.8 | 48.3 | 46.5 | 39.6 | 28.7 | 30.3 |
| +DMGAN(Link + Degree) | 50.2 | 48.6 | 47.0 | 41.2 | 33.3 | 33.7 |
| +DPMoE+DMGAN(Link) | 52.2 | 51.6 | 51.2 | 50.4 | 49.5 | 47.6 |
| +DPMoE+DMGAN(Link+Degree) | **52.7** | **52.0** | **51.8** | **51.2** | **50.3** | **48.8** |

# E  DEFENSE AGAINST GRAPH MODIFICATION ATTACKS

In the main section of the paper, we mainly present the performance of DRAGON under graph injection attacks. To evaluate the effectiveness of DRAGON under graph modification attacks, we apply GR-BCD and BR-BCD attacks (Geisler et al., 2021) on our method and baselines, including SOTAs (Soft-Medoid/Soft-Median GDCs (Geisler et al., 2021) and GAME). The results are summarized below in Table 14.

Table 14: The robust accuracy of Soft-Median GDC and Soft-Medoid GDC without or with our GAME framework on the Cora dataset with the global attacks (GR-BCD & PR-BCD, $\epsilon = 0.1$) proposed by Soft-Medoid GDC. We set the number of experts to 10 and the hidden units of each expert to 32. We run them on three random splits and report the mean and standard error results.

| | GR-BCD | PR-BCD |
|---|---|---|
| GCN | 0.622 ± 0.003 | 0.645 ± 0.002 |
| GDC | 0.677 ± 0.005 | 0.674 ± 0.004 |
| PPRGo | 0.726 ± 0.002 | 0.700 ± 0.002 |
| SVD GCN | 0.755 ± 0.006 | 0.724 ± 0.006 |
| Jaccard GCN | 0.664 ± 0.001 | 0.667 ± 0.003 |
| RGCN | 0.665 ± 0.005 | 0.664 ± 0.004 |
| Soft-Median GDC | 0.765 ± 0.001 | 0.752 ± 0.002 |
| Soft-Medoid GDC | 0.775 ± 0.003 | 0.761 ± 0.003 |
| Soft-Median GDC (+GAME) | 0.772 ± 0.005 | 0.759 ± 0.005 |
| Soft-Medoid GDC (+GAME) | 0.780 ± 0.007 | 0.772 ± 0.006 |
| DRAGON | 0.788 ± 0.012 | 0.784 ± 0.007 |

# F  DEFENSE AGAINST GRAPH ADAPTIVE ATTACK

In our paper, we primarily focus on more advanced and scalable injection attacks involving malicious node features and edges. On the other hand, adaptive attacks LowBlow (Entezari et al., 2020) and the method from Mujkanovic et al. (2022) primarily focus on structure (edge) perturbations and are usually limited to small graphs due to their cubic complexity in computing the full set of adjacency eigenpairs. Consequently, adaptive attacks fail to scale to medium-sized graph datasets like Flickr and large graph datasets like Reddit. Additionally, they cannot be easily scaled with higher intensity, which is essential for our investigation into the emergence of robustness degradation. Moreover, our DMGAN module is non-differentiable during graph reconstruction, making adaptive attacks difficult to optimize a specific loss function on our model. Table 15 provides a referential comparison when models are attacked by adaptive attacks.

Table 15: The robust performance of our method under adaptive attacks on the Cora dataset.

| Model | Vanilla | SVD | ProGNN | GNNGuard | GARNET | Soft-Median-GDC | Ours |
|---|---|---|---|---|---|---|---|
| Entezari et al. (2020) | 74.77 ± 0.71 | 26.03 ± 2.76 | 69.88 ± 1.61 | 74.80 ± 0.95 | 77.71 ± 0.95 | 77.62 ± 0.80 | **80.28 ± 1.08** |
| Mujkanovic et al. (2022) | 75.32 ± 0.71 | 20.11 ± 3.32 | 74.87 ± 2.02 | 72.53 ± 0.25 | 75.27 ± 0.74 | 77.15 ± 0.56 | **79.92 ± 1.35** |

## G    EXPLORATION OF ALTERNATIVE GRAPH ROBUST LEARNING METHODS (SLIGHTLY OUTSIDE THE SCOPE OF OUR PAPER)

In Appendix E and Appendix F, we have assessed several of these baselines under different types of attacks, furnishing a comprehensive discussion along with experimental results. Therefore, here, we aim to elucidate the rationale behind not selecting some recent methods as our primary baselines.

Many methods are only effective for specific attacks with ad hoc designs, making the results less generalizable and practical. SoftMedian (Geisler et al., 2021) and GARNET (Deng et al., 2022) are only designed to defend against structure (edge) perturbations between original nodes by cleaning the structure, and GCN-Jaccard (Wu et al., 2019b) uses structure-based preprocessing. Although structure-based solutions handle structure perturbations well and can scale to large graphs by considering only structure, they fail to defend against advanced node injection attacks with both malicious features and edges. Table 16 shows that the performances of these structure-based methods are less robust than GAT with adversarial training and even close to vanilla GAT under node injection attacks. This structure-based weakness has been discussed in the benchmark (Zheng et al., 2021), which also avoids them as baselines.

Current graph anomaly detection methods still face the accuracy problem when used to remove malicious nodes, i.e., they falsely identify too many original nodes as anomaly nodes. The recent method CoLA (Liu et al., 2021) provides a detection performance of 0.751 on the Flickr dataset, which means that when using it to remove attacks, it may introduce even more extra node noise (0.249) than the highest intensity attacks (0.056). Table 16 shows that anomaly detection cannot be easily generalized to our scenario, i.e., the emergence of robustness degradation.

Table 16: The performance of DRAGON and other methods under FGSM attack of varying intensities on the Flickr dataset. GAD is a representative **graph anomaly detection** method and Struc. represents methods are only designed for graph **structure** attacks or only defend in structural ways.

| Methods | nulla | I | II | III | IV | V |
|---|---|---|---|---|---|---|
| Vanilla GAT | 50.0 | 47.8 | 44.1 | 34.2 | 24.5 | 24.9 |
| CoLA (GAD *Anomaly Detection*) | 38.2 | 25.7 | 19.8 | 17.6 | 14.5 | 15.2 |
| GCN-Jaccard (Struc.) | 47.3 | 41.2 | 33.6 | 18.3 | 14.8 | 15.4 |
| SoftMedian (Struc.) | 51.0 | 48.3 | 46.2 | 39.8 | 27.2 | 28.7 |
| GARNET (Struc.) | 51.8 | 49.9 | 42.1 | 35.6 | 29.2 | 30.1 |
| GAT+AT | 44.2 | 44.3 | 44.0 | 43.1 | 42.8 | 42.4 |
| Ours | **52.7** | **52.0** | **51.8** | **51.2** | **50.3** | **48.8** |

## H    COMPLEXITIES

Here we analyze the time and memory complexities of our method. Let $N$ be the number of nodes, $N'$ be the number of nodes after masking, $E$ be the number of edges, $E'$ be the number of edges after masking, $d$ be the size of the hidden channels (we assume it is of the same order as the size of the input features), $l_1$ be the number of encoder layers in the DMGAN, $l_2$ be the number of MLP layers in the DMGAN decoder, $l_3$ be the number of DPMoE layers, $k$ be the number of experts, our comprehensive complexity analysis is structured as follows:

**DMGAN Module.** DMGAN encoder part obtains encoded representations with time complexity $O(l_1 d^2 N')$ and memory complexity $O(l_1 dN' + l_1 d^2)$. - The DMGAN MLP decoder part decodes the degree regression with time complexity $O(l_2 d^2 N')$ and memory complexity $O(l_2 dN' + l_2 d^2)$. MLP also predicts whether edges exist or not by decoding $2E'$ times node feature element-wise product, with $O(2l_2 d^3 E')$ time and $O(2l_2 dE' + l_2 d^2)$ memory complexity. - The total time complexity of the DMGAN module is $O(l_1 d^2 N' + l_2 d^2 (N' + 2dE'))$, and the memory complexity is $O(l_1 d(N' + d) + l_2 d(N' + 2E' + 2d))$.

**DPMoE Module.** In one layer DPMoE, $N/k$ nodes are distributed into each expert, resulting in $O(d^2 N/k)$ time and $O(d^2 + dN/k)$ memory complexity. For all $k$ experts, the total time complexity is $O(d^2 N)$ and memory complexity is $O(kd^2 + dN)$ - Gating network is trained with $O(dkN)$ time complexity and constant $O(dk)$ memory complexity for gating in one layer. - Overall time complexity of DPMoE module is $O(l_3 dN(d + k))$ and memory complexity is $O(l_3 d(N + kd + k))$.

**DRAGON (Complete Model).** Overall time complexity of DRAGON is $O(l_1 d^2 N' + l_2 d^2 (N + 2dE') + (l_3 dN(d + k))$, and memory complexity is $O(l_1 d N(N' + d) + l_2 d(N' + 2E' + 2d) + l_3 d(N + kd + k))$.

Given these additions and clarifications, when constants are stripped away (i.e., mask rate), the complexities for our methodology remain linear (i.e., $O(N+N'+E')$), matching our earlier simplified analysis and empirical outcomes. Empirical assessments, as presented in Table 1, underscore that most baselines with $O(N^2)$ complexities (e.g., GCNSVD and GNNGuard) find scalability challenging, especially with medium to large graphs. This contrasts with our linear complexity method which demonstrates better scalability. For brevity, the comparison between some representative baselines and our method is listed in Table 17. In addition, comparing the standard training time, DRAGON shows better performance and less training time compared to GNNGuard and GCNSVD since they have a significantly higher time complexity of $O(N^2)$. The details of training time are listed in Table 18.

Table 17: Time and Memory Complexity of representative graph robust learning methods.

| Complexity | RGCN | GAME | EvenNet | GCNSVD | GNNGuard | Ours |
|---|---|---|---|---|---|---|
| Time | $O(E)$ | $O(N + E)$ | $O(E)$ | $O(N^2 + E)$ | $O(E)$ | $O(N + E)$ |
| Memory | $O(E)$ | $O(N)$ | $O(N + E)$ | $O(N^2)$ | $O(N^2)$ | $O(N + E)$ |

Table 18: The training times for models on the Cora dataset.

| Model Name | RGCN | GAT | EvenNet | GAME | DRAGON | GNNGuard | GCNSVD |
|---|---|---|---|---|---|---|---|
| Training Time (s) | 00:03 | 00:04 | 00:11 | 00:18 | 00:23 | 01:03 | 05:28 |

# I  A BITTER LESSON FROM THE REAL WORLD: MASSIVE NODE INJECTION ATTACK IS POSSIBLE

**Learning from the highly corrupted graph is important.** In representation learning, the ability to derive effective representations from highly corrupted data has been a subject of great interest. This pursuit is evident in various domains, including computer vision, where the goal is to recover original images from heavily noisy pictures, and natural language processing, where the challenge is to learn useful language representations from low-quality Internet corpora. Similarly, in the context of graphs, the significance of learning effective representations from highly corrupted data cannot be overstated. Therefore, the ability to learn high-quality representations from highly noisy Internet social graphs is of immense practical importance.

**Massive node injection on the graph is possible.** The notion of "unnoticeability" or "imperceptibility" is indeed critical in general adversarial attacks, which constrains the attack intensity. However, transposing this concept directly from computer vision to graph scenarios brings about its own set of challenges, which allows strong attack intensity to be realistic on graphs:

*(i) Ill-definition in Graph Scenarios:* As highlighted by research on adversarial learning on non-Euclidean data, specifically on graphs (Zheng et al., 2021):

- *"However, this assumption is controversial: If defenders have the original graph, they can simply train the model on that one; If defenders do not have the original graph (the general case for data poisoning where defenders can not tell whether the data are benign or not), then it does not make sense to keep unnoticeability. This is different from the case of images, where unnoticeability can be easily judged by humans even without ground-truth images."*

- *"The attackers may perturb the graph structure or attributes within the scope of unnoticeability defined by themselves, while defenders have to depend on their own observations to discover. However, even if defenders notice that the degree distribution is different, it is still hard to identify specific malicious nodes or edges from the entire graph."*

- *"Thus, we do not add too many constraints and we insist that the notion like "unnoticeability" will be refined during the arms race between attackers and defenders. For example, if an advanced*

*defense proposes a measure to identify malicious nodes with high probability, then the attackers can decide by themselves to refine the constraints based on this measure."*

*(ii) Dynamics of Real-world Graphs:* Networks like those on the internet undergo constant modifications, often through the addition of new nodes. This continual change provides a conducive environment for malicious node insertions, often without the constraints of "unnoticeability". Identifying these alterations, especially the specific malicious nodes and edges, is challenging, as highlighted in Global-Response Section 3.

*(iii) High-Intensity Attacks in Reality:* Contrary to the notion of limited attack budgets, real-world scenarios, especially on Internet social platforms, have demonstrated intense attacks. For instance, media outlets have reported massive bot influxes on platforms like Twitter. These bots, operating at high intensities, spread disinformation on an unprecedented scale, and interact with other users without being detected (same with injection attacks). One real-world report[1] have shown that bots present on these platforms at high intensities (20%). In such scenarios, where attacks are noticeable, classifying new nodes on attacked graphs remains a crucial challenge. The societal concerns derived from social media are always at the forefront of machine learning research (Vosoughi et al., 2016, 2017, 2018; Liu et al., 2024).

In conclusion, our work primarily emphasizes high-intensity attack scenarios, given their growing relevance in real-world graph contexts. These scenarios differ significantly from traditional attack situations in vision or language domains. Our approach, designed to address the emergence of severe robustness degradation, gains notable importance in these graph-related contexts.

---

[1] Twitter Bots Poised to Spread Disinformation Before Election, *The New York Times, October 29, 2020.*

