# OpenReview forum: "Mitigating Emergent Robustness Degradation while Scaling Graph Learning"
_ICLR.cc/2024/Conference — ICLR 2024 poster_

### Official Review · Reviewer_Fsb2 · 2023-10-24

**Soundness:** 3 good
**Presentation:** 2 fair
**Contribution:** 2 fair
**Rating:** 6
**Confidence:** 4

**Summary:**

This paper proposes a framework DRAGON that is robust against Graph Injection Attack.
The framework first adopts an edge denoising module implemented by an auto-encoder.
Then, a Differentially-Private Mixture-of-Experts (DPMOE) layer is used as a robust Graph Convolution layer.
Extensive experiements verify the effectiveness and efficiency of DRAGON.

**Strengths:**

1. The proposed framework holds siginificant perfromance superiority over defense baselines against various types of attacks.
2. The proposed framework is scalable to large graphs, like AMiner with more than 600,000 nodes.

**Weaknesses:**

1. The module of DRAGON is not first proposed in the paper, which slightly limits the novelty of the proposed framework.
2. It would be better if the presentation is more clear. For example, the notations in the lemma are not well introduced. The setting of the attack and defense is not included as well.
3. The whole framework is quite complex and not in an end-to-end manner.
It includes quite a few hyperparameters, and choosing a propoer GNN backbone seems to matter, which would be a concern when applied to real-world scenarios.

Other concerns are listed in Questions.

**Questions:**

1. How is the auto-encoder trained? Is it trained on the clean graph or trained on the attacked graph?
If the denoising module is trained on the attacked graph, why auto-encoder enjoys such a good performance in recovering clean graph structrue?
Including more discussion about the intuition why the denosing part is effective would be better.
2. What the reuslts would be if the denoising module is compiled with other defense model?  For example, GATGuard + DMGAN.
It looks like the two modules of DRAGON can be decoupled.
3. It is amazing that DRAGON is so efficient. Dose the reported time in Table 18 include the training time of DMGAN?
4. DRAGON adopts other defense model as backbone, which could seems a little unfair. I wonder how DRAGON would perform when compiled with other basic GNNs like GCN? Is it sill competitive?
5. Why the baselines like GATGuard and EvenNet not coupled with AT?

---

> ### Author Response · Authors · 2023-11-18
> **Response to Reviewer Fsb2 [1/4]**
>
> Dear Reviewer Fsb2:
>
> Thanks for your insightful review. Below, we aim to provide clarifications. We have integrated all updates into the revision of our paper.
>
> ---
> **[W1. The module of DRAGON is not first proposed in the paper, which slightly limits the novelty of the proposed framework.]**
>
> - **Novelty on DMGAN** Although multiple graph masked auto-encoders [1, 2, 3] have been proposed, the intrinsic denoise ability of graph masked auto-encoder is first revealed by this paper. We are the first attempt to design a denoise graph masked auto-encoder and corresponding pretraining-recovery paradigm to denoise the attacked graph.
>
> - **Novelty on DPMoE** Previous research on ensemble robustness [10, 11, 12] and MoE techniques [13, 14] only utilize the common components, and each of them is the same as others. We first assign DP noises with different magnitudes to each expert to counteract attacks at different intensities by gate routing. Furthermore, we provide the theoretical robustness guarantee for our DP-based MoE, which to the best of our knowledge has not been discussed in any ensemble robustness or MoE paper before.
>
> - **Novelty on motivation and setting** Although DP, MoE, and MAE have been proposed in other domains, we identify their particular properties. Specifically, we find (i) masked graph auto-encoder in our design is able to recover clean graphs even under high-intensity attacks; (ii) matching DP noise magnitudes can help model better defense attacks with different intensities; (iii) MoE can select the most matching DP expert to handle the attack with specific intensity. Then, to solve this unique and realistic task (graph learning with anti-degradation robustness against increasing-intensity attacks), we deftly utilize these "particular properties" to design well-motivated new model architecture (DPMoE) and denoising paradigm (DMGAN). Under attacks with different intensities, DMGAN is always able to reconstruct clean graphs and DMGAN selects the most matching DP expert to defense attacks with different intensities.
>
>
> Finally, our framework is the first to successfully address the phenomenon of severe robustness degradation under increasing intensity graph attacks:
> > Many existing defense methods, such as RGCN and GNN-SVD, effectively handle mild node injection attacks but experience an emergent decline in robustness as attack intensity increases. As shown in Figure 1, when attack intensity surpasses a threshold of 300 injected nodes, error rates for many models surge by more
> than 50%. _(Quoted from Introduction Section in our submission)_
>
> We extensively test DRAGON on various graph datasets, where it not only outperforms popular baselines but also represents the first successful attempt to avoid severe robustness degradation phenomena (which are widespread in the real world, as discussed in Appendix I) while maintaining scalability.
>
> ---
> **[W2. It would be better if the presentation is more clear. For example, the notations in the lemma are not well introduced. The setting of the attack and defense is not included as well.]**
>
> We appreciate your feedback!
> - Recognizing the need for better clarity, we have revised our manuscript to reintroduce and clearly define the notations at the start of the lemma, even if they have been previously mentioned.
>
> - To further clarify the attack and defense settings, we have now included detailed descriptions in Appendix C.

---

> ### Author Response · Authors · 2023-11-18
> **Response to Reviewer Fsb2 [2/4]**
>
> **[W3. The whole framework is quite complex and not in an end-to-end manner. It includes quite a few hyperparameters, and choosing a proper GNN backbone seems to matter, which would be a concern when applied to real-world scenarios.]**
>
> We appreciate your comment and would like to clarify the following points:
> - Our framework adopts a two-stage design (denoising -> prediction). While this may seem complex in concept, it does not translate into complexity during real implementation or inference. In particular, our experimental results show that the training and inference processes remain efficient (not overly complex compared to baselines). Moreover, the two-stage approach is a proven strategy/philosophy in the domains of vision/language processing for stable learning in the real world, as evidenced by studies [8, 9, 15, 16].
>
> - DRAGON with different GNN backbones has different performance, but DRAGON uniformly improves backbone robustness a lot. We further provide the performance of DRAGON with GCN backbone in Table F and Table G. Although DRAGON with GCN backbone is slightly less robust than DRAGON with GAT backbone in the paper, it still outperforms other baselines.
> - We analyze the sensitivity of 5 key hyperparameters (3 for DMGAN, 2 for DPMoE) in Tables 6, 12, Figures 9, and 10, respectively. According to the analysis of tables and figures, these hyperparameters are not very sensitive to downstream performance and they are also easy to tune according to the observed regular pattern from different scales' experiments, then reduce the tuning complexity in real-world datasets.
>
> ----
> **[Q1. How is the auto-encoder trained and is it trained on the clean graph or trained on the attacked graph? If the denoising module is trained on the attacked graph, why auto-encoder enjoy such a good performance in recovering clean graph structure? Including more discussion about the intuition why the denoising part is effective would be better.]**
>
> - We train the DMGAN on the clean training graph, which follows the settings (dataset configurations, evaluation configurations, etc.) of the graph robustness benchmark [4].
>
> - The path masking strategy disrupts the edges between consecutive nodes, prompting the model to grasp the structured dependency and higher-order proximity inherent in node relationships from 'mask noise'. This, in turn, enables the model to reconstruct a clean graph from the initially attacked one.
>
> - The initial intuition of this design is inspired by vision representation research [6], where researchers observed a phenomenon: reconstructed examples, although distinct from the ground truth, remained semantically plausible. Therefore, we propose DMGAN to effectively remove edges that lack semantic plausibility and deviate from the normative distribution of structural characteristics.

---

> ### Author Response · Authors · 2023-11-18
> **Response to Reviewer Fsb2 [3/4]**
>
> ---
> **[Q2. What the results would be if the denoising module is compiled with other defense models? For example, GATGuard + DMGAN. It looks like the two modules of DRAGON can be decoupled.]**
>
> DMGAN can also be combined with other defense models and improve the robustness performance: in Table D below, GATGuard + DMGAN has better robustness performance than GATGuard. We also combine DMGAN with another competitive baseline, GAME, and show the results in Table E, where GAME+DMGAN also has better robustness than GAME.
>
> Table D. The robustness performance of GATGuard and GATGuard + DMGAN under FGSM attack on the Cora dataset.
> |                | Cora. nulla | Cora. I  | Cora. II | Cora. III | Cora. IV | Cora. V  | Flickr. nulla | Flickr. I | Flickr. II | Flickr. III | Flickr. IV | Flickr. V |
> |----------------|-------------|----------|----------|-----------|----------|----------|---------------|-----------|------------|-------------|------------|-----------|
> | GATGuard       | **81.7**    | 80.6     | 80.7     | 80.7      | 79.4     | 79.4     | OOM           | OOM       | OOM        | OOM         | OOM        | OOM       |
> | GATGuard+DMGAN | 81.6        | **81.0**| **80.8**| **80.8**  | **80.3**  | **80.1** | OOM           | OOM       | OOM        | OOM         | OOM        | OOM       |
>
>
> Table E. The robustness performance of GAME and GAME + DMGAN under FGSM attack on the Cora dataset.
> |                | Cora. nulla | Cora. I  | Cora. II | Cora. III | Cora. IV | Cora. V  | Flickr. nulla | Flickr. I | Flickr. II | Flickr. III | Flickr. IV | Flickr. V |
> |----------------|-------------|----------|----------|-----------|----------|----------|---------------|-----------|------------|-------------|------------|-----------|
> | GAME           | **85.5**    | 81.7     | 80.1     | 79.1      | 78.3     | 77.3     | **52.2**      | 45.6      | 43.0       | 41.1        | 40.8       | 38.4      |
> | GAME+DMGAN     | 85.2        | **81.8** | **81.2** | **80.4**  | **78.6** | **77.9** | 52.0          | **46.7**  | **43.3**   | **43.0**    | **41.5**   | **39.8**  |
>
> As shown in the previous Table D and Table E, for the most competitive baselines, GATGuard + DMGAN still suffers from out-of-memory on GPU and GAME + DMGAN suffers from severe robustness degradation. These results again reflect the scalability and robustness against degradation of DPMoE.
>
>
> ---
> **[Q3. Does the reported time in Table 18 include the training time of DMGAN?]**
>
> Yes. Our design is efficient for two reasons: (i) according to the analysis in Appendix H, the time complexity is linear: O(N+E), which indicates training time is **linearly** increasing with the number of node N and the number of edge E; (ii) although DPMoE is complex and consists of multiple experts, it dispatches nodes to experts without repeat training on the same data. This efficient property has been analyzed in previous research [7].

---

> ### Author Response · Authors · 2023-11-18
> **Response to Reviewer Fsb2 [4/4]**
>
> **[Q4. How DRAGON would perform when compiled with other basic GNNs like GCN? Is it still competitive?]**
>
> DRAGON serves as a plug-and-play framework with the flexibility to be integrated with different GNN backbones, i.e., the motivation of DRAGON is the generality of helping the defense model/backbone. We plug DRAGON into the basic GAT on most datasets and only with GATGuard (w.o. AT) on small datasets. Here we plug DRAGON into GCN on several datasets with different scales in Tables F and G, which also show the competitive performance and scalability of our method. Although DRAGON with GCN backbone performs less robustly on CiteSeer, DRAGON+AT still leads in robustness.
>
> Table F. The robustness performance of our method with GCN backbone under FGSM attacks on the Citeseer (small-scale) dataset.
> | methods  | nulla |   I  |  II  |  III |  IV  |   V  |
> |-------------------|:-----:|:----:|:----:|:----:|:----:|:----:|
> | GATGuard|  72.8 | 72.5 | 72.4 | 72.5 | 72.4 | 72.5 |
> | GAME+AT | **76.1** | 72.4 | 69.3 | 66.7 | 64.6 | 62.8 |
> | Ours    | 73.7  | 72.3 | 70.4 | 69.4 | 69.2 | 67.9 |
> | Ours+AT | 73.4  | **73.1** | **73.0** | **72.8** | **72.8** | **72.6** |
>
> Table G. The robustness performance of our method with GCN backbone under FGSM attacks on Flickr (medium-scale) and Reddit (large-scale) datasets. The improvement denotes the improvement of our method compared to *the best performance among all baselines*.
> | Dataset & Training  | nulla |   I  |  II  |  III |  IV  |   V  |
> |-------------------|:-----:|:----:|:----:|:----:|:----:|:----:|
> | Flickr, Ours      |  52.5 | 51.3 | 50.7 | 49.0 | 47.4 | 46.8 |
> | Flickr, Ours: improvement |  +1.7 | +3.1 | +4.9 | +1.5 | +3.9 | +3.4 |
> | Flickr, Ours+AT   |  52.1 | 51.8 | 51.2 | 49.7 | 48.2 | 47.4 |
> | Flickr, Ours+AT: improvement |  +2.2 | +3.6 | +5.6 | +2.2 | +4.8 | +4.0 |
> | Reddit, Ours      |  95.8 | 95.6 | 95.5 | 95.4 | 95.3 | 95.4 |
> | Reddit, Ours: improvement |  +0.0 | +0.0 | +0.1 | +0.1 | +0.1 | +0.2 |
> | Reddit, Ours+AT   |  95.7 | 95.6 | 95.6 | 95.6 | 95.5 | 95.5 |
> | Reddit, Ours+AT: improvement |  -0.1 | +0.0 | +0.2 | +0.3 | +0.3 | +0.3 |
>
>
> ---
> **[Q5. Why the baselines like GATGuard and EvenNet not coupled with AT?]**
>
> As shown in Table H, training GATGuard and Evennet with AT hurts their robustness. Therefore, we don't combine GATGuard and Evennet with AT since they are stronger baselines without AT for fair comparisons.
>
> Table H. The performance of baselines under FGSM attack on the Cora dataset.
> |             | nulla    | I        | II       | III      | IV       | V        |
> |-------------|----------|----------|----------|----------|----------|----------|
> | Evennet     | 82.4     | 67.7     | 53.3     | 39.6     | 28.9     | 23.8     |
> | Evennet+AT  | 82.6     | 64.4     | 55.1     | 32.7     | 24.5     | 21.5     |
> | GATGuard    | 81.7     | 80.6     | 80.7     | 80.7     | 79.4     | 79.4     |
> | GATGuard+AT | 80.3     | 79.2     | 79.2     | 79.0     | 78.2     | 78.3     |
> | Ours        | **84.1** | **84.2** | **84.1** | **84.0** | **84.0** | **83.3** |
>
> It is worth noting that not all defense methods can improve the robustness via adversarial training. Previous benchmark research [4] also has shown that GATGuard+AT has less robustness compared with GATGuard. This phenomenon can be explained by the special architecture designed against modification attacks, and introducing adversarial samples via max-min adversarial training often instigates a shift in data distribution compared to real-world adversarial samples, a phenomenon discussed at length in [5].
>
> ---
> **References:**
>
> [1] GraphMAE: self-supervised masked graph autoencoders, KDD 2022
>
> [2] Heterogeneous graph masked autoencoders, AAAI 2022
>
> [3] S2GAE: self-supervised graph autoencoders are generalizable learners with graph masking, WSDM 2023
>
> [4] Graph robustness benchmark: Benchmarking the adversarial robustness of graph machine learning, NeurIPS 2021
>
> [5] Adversarial examples improve image recognition, CVPR 2020
>
> [6] Masked autoencoders are scalable vision learners, CVPR 2022
>
> [7] Switch transformers: scaling to trillion parameter models with simple and efficient sparsity, JMLR 2022.
>
> [8] Detecting Textual Adversarial Examples through Randomized Substitution and Vote, UAI 2022
>
> [9] Combating Adversarial Misspellings with Robust Word Recognition, ACL 2019
>
> [10] TRS: Transferability reduced ensemble via promoting gradient diversity and model smoothness, NeurIPS 2021
>
> [11] Building robust ensembles via margin boosting, ICML 2022
>
> [12] Diversifying vulnerabilities for enhanced robust generation of ensembles, NeurIPS 2020
>
> [13] Outrageously large neural networks: The sparsely-gated Mixture-of-Experts layer, ICLR 2017
>
> [14] Switch transformers: Scaling to trillion parameter models with simple and efficient sparsity, JMLR 2022
>
> [15] An efficient image compression model to defend adversarial examples, CVPR 2019
>
> [16] Deflecting adversarial attacks with pixel deflection, CVPR 2018

---

> ### Comment · Reviewer_Fsb2 · 2023-11-19
> **Questions about the evaluation setting.**
>
> Thanks for the authors' detailed review. Most of my concerns have been well addressed, but I'm still worried that "train the DMGAN on the clean training graph" is problematic.
>
> To the best of my knowledge, although the defending model is trained on the clean graph during evaluation, it is not allowed for the defender to access or "remember" the clean graph in their model design.
> Using DMGAN to reconstruct the clean graph where it is trained seems to violate the rule.
> As it said in "Graph Robustness Benchmark":
> *For defenders: (a) They have knowledge about the graph excluding the test nodes to be attacked. (b) They are allowed to use any method to increase the adversarial robustness, but do not have prior knowledge about the edges/nodes that are modified/injected.*
>
> If the clean graph is obtained by the defender, the edges/nodes modified/injected can be detected, which I think can not be called following the setting of the graph robustness benchmark.
> It would be better for the authors to clarify the problem setting or train DMGAN on the attacked graph.
>
> And a follow-up weakness seems to be that DRAGON can not tackle poison attacks, where the whole model needs to be trained on the attacked graph.

---

> ### Author Response · Authors · 2023-11-19
> **Thank you for your continued engagement and feedback!**
>
> We appreciate your concern regarding the training process of DMGAN and its adherence to the settings of the Graph Robustness Benchmark (GRB) and adversarial training. We assure that our framework (including the DMGAN module), strictly complies with these settings during both the training and evaluation phases. To clarify, here is a more detailed explanation of our process aligned with GRB guidelines:
>
> - **Training Stage**: During training, DMGAN only has access to the **clean training graph _on the training set_, as the model's input**. This input is subsequently subjected to adversarial attacks (e.g., PGD attack) to generate adversarial training examples, following standard adversarial training practices.
>
> - **Evaluation Stage**: In test evaluation, DMGAN operates only on the **test graphs**, which are also attacked (e.g., PGD attack). It does not have access to, nor does it recall, any clean test graph information. Hence, there is no test-set information leakage concerning the test nodes targeted for attack or the modifications/injections made to the edges/nodes.
>
> Based on training data, DMGAN is trained to learn general patterns and representations that enable the reconstruction of a perturbed graph into a cleaner version, _using only the data from the training set_ and instead of using data from the test set. During test evaluation, given only an attacked test graph from the test set as input, it attempts to recover a clean(er) graph **without** any prior information leakage about the clean information from the test graph set. Therefore, DMGAN adheres to the GRB's conditions, which prohibit access to the clean graph during evaluation and any information leakage/foreknowledge of attack modifications.
>
> In summary, our experimental setup uses the training and test data provided by the GRB. We train DMGAN on the **clean training-set graph** and apply it to the **attacked test-set graph**. So our approach is fully compliant with the GRB rules: DMGAN does not have access to the clean test graph and lacks prior information about the specific edges/nodes that have been modified or injected.
>
> In fact, DMGAN's strategy (i.e., firstly cleaning the input test data and then making robust test predicting easier for evasion defense) is also a widely used philosophy in robust vision/language processing for evasion defense, as evidenced by studies [5, 6, 7, 8]: For example, robust NLP methods referenced in [5, 6] similarly transform noisy or attacked test text into a cleaner state without access to the original clean information of test data as a first step (akin to our DMGAN approach), then simplifying the task of making robust predictions in an evasion defense context as a second step. Thus, they also do not break the rule, which is also consistent with the basic rules of adversarial robustness summarized by the Graph Robustness Benchmark.
>
> ---
>
> We would also like to clarify that the poising attack is a different setting from our paper's evasion attack. As GRB's said:
> > _Poisoning: Attackers generate corrupted graph data and assume that the targeted model is trained on these data to get a worse model (usually worse on clean test data)._
> >
> > _Evasion: The target model has already been trained, and (during the test phase) attackers can generate corrupted test graph data to affect its inference._
>
> Our design mainly concentrates on defending against evasion attacks provided by GRB, so defending against poisoning attacks is out of our scope. Since there is already a lot of research [2, 3, 4, 9] (only) focusing on attack-defense under graph evasion attack/defense setting, and evasion defense is also considered more realistic than poisoning setting in GRB [2], we would like to humbly clarify our targeted exploration of evasion attack/defense settings, while not encompassing poisoning attacks, doesn't harm the core contribution of our work.
>
> ___
> References:
>
> [1] Towards Deep Learning Models Resistant to Adversarial Attacks, ICLR 2018
>
> [2] Graph Robustness Benchmark: Benchmarking the Adversarial Robustness of Graph Machine Learning, NeurIPS 2021
>
> [3] Chasing All-Round Graph Representation Robustness: Model, training, and Optimization, ICLR 2023
>
> [4] Understanding and Improving Graph Injection Attack by Promoting Unnoticeability, ICLR 2022
>
> [5] Detecting Textual Adversarial Examples through Randomized Substitution and Vote, UAI 2022
>
> [6] Combating Adversarial Misspellings with Robust Word Recognition, ACL 2019
>
> [7] An Efficient Image Compression Model to Defend Adversarial Examples, CVPR 2019
>
> [8] Deflecting Adversarial Attacks with Pixel Deflection, CVPR 2018
>
> [9] Tdgia: Effective Injection Attacks on Graph Neural Networks, KDD 2021

---

> > ### Comment · Reviewer_Fsb2 · 2023-11-19
> > **Thanks for your detailed clarification.**
> >
> > Thanks for your detailed clarification.
> > I have a few further questions.
> >
> > * During training, is the MAE trained on the subgraph consisting of only training nodes?
> > If the training set is relatively smaller, the subgraph would be of poor connectivity and inaccurate degree information.
> > I wonder if it would be a limitation for DRAGON when applied to large and low label-rate settings.
> > * During the evaluation, does the decoder only remove edges?
> > And the edges between clean test nodes and train nodes are possible to be removed as well?
> >
> > Again, thanks for the authors' efforts in explaining the setting and technical details.

---

> > > ### Author Response · Authors · 2023-11-20
> > > **Thank Review Fsb2 for acknowledging our previous clarifications and for raising further insightful questions.**
> > >
> > > We welcome the opportunity to delve deeper into the intricacies of the MAE's training dynamics and the decoder's role during test evaluation.
> > >
> > > ---
> > > **[Q1: Training on Subgraphs and Impact on DRAGON in Large, Low Label-Rate Settings]**
> > >
> > > Yes, DMGAN (which can also be read as MAE) is trained exclusively on the subgraph composed of training nodes. Consistent with the GRB settings, we use a default **training split of 60%.**
> > >
> > > To explore how high and low label rate settings affect DRAGON, we experimented with varying training set splits, as detailed in Table H:
> > >
> > > _Table H. DRAGON's downstream performance (classification accuracy) across different training set splits on the Flickr dataset under the FGSM attack._
> > >
> > > | Training set split | nulla    | I        | II       | III      | IV       | V        |
> > > |--------------------|----------|----------|----------|----------|----------|----------|
> > > | 0.8                | **52.9** | **52.1** | **52.1** | 51.1     | 50.2     | **49.2** |
> > > | 0.6                | 52.7     | 52.0     | 51.8     | **51.2** | **50.3** | 48.8     |
> > > | 0.4                | 52.3     | 52.1     | 51.5     | 49.8     | 49.7     | 49.4     |
> > > | 0.2                | 51.9     | 51.6     | 51.1     | 49.9     | 49.0     | 48.5     |
> > >
> > > While there is a nominal drop in overall performance as the number of training nodes decreases, DRAGON maintains robustness by:
> > >
> > > (i) when the training set is too small, the affect of weak subgraph connectivity and degree loss information is mitigated by the predominance of the link loss term (analyzed in the fine-grained ablation study of Table 13);
> > >
> > > (ii) DMGAN's denoising function is an extra bonus to DRAGON's core robustness, i.e., the overall robustness of the framework is primarily strengthened by the DPMoE module, which is trained to directly withstand adversarial inputs.
> > >
> > > Therefore, our DRAGON's overall framework maintains stable robustness and performance under the affect of large and low label rate settings.
> > >
> > > ---
> > > **[Q2: During the evaluation, does the decoder only remove edges? And the edges between clean test nodes and train nodes are possible to be removed as well?]**
> > >
> > > Yes, DMGAN only removes edges, which is by determining whether to keep or remove the edge (considering the setting is node injection attack, DMGAN also doesn't add/predict non-existent edges into attacked input).
> > > This includes potential removals between clean nodes, though such instances remain proportionately low.
> > > Here we provide a detailed analysis:
> > >
> > > Let's denote TPR-N↑ (FNR-N↑) as TPR (FNR) for link prediction between normal/original nodes; and TPR-A↑ (FNR-A↑) as TPR (FNR) for link prediction between original nodes and injection/attack nodes. Again, since DMGAN doesn't need to predict non-existent edges on the attacked graph (considering the setting is node injection attack), we don't have FPRs and TNRs. An analysis of true positive rates (TPR) and false negative rates (FNR) is provided in Table I:
> > >
> > > _Table I. TPRs and FNRs of DMGAN under FGSM attack on the Cora dataset._
> > > |        | nulla | I     | II    | III   | IV    | V     |
> > > |--------|-------|-------|-------|-------|-------|-------|
> > > | TPR-N↑ | 0.954 | 0.941 | 0.935 | 0.931 | 0.929 | 0.926 |
> > > | FNR-N↓ | 0.046 | 0.059 | 0.065 | 0.069 | 0.071 | 0.074 |
> > > | TPR-A↓ | N.A.  | 0.638 | 0.674 | 0.691 | 0.724 | 0.749 |
> > > | FNR-A↑ | N.A.  | 0.362 | 0.326 | 0.309 | 0.276 | 0.251 |
> > >
> > > FNR-N (less than 7.4%) indicates the ratios of edges between original nodes are wrongly removed by DMGAN. Combine with the ablation study in Table 2 and Table 13, we find that such extra wrong removal introduced by DMGAN can only lead to
> > >
> > > (i) 0.1% performance decrease on Cora and CiteeSeer datasets of DRAGON's final output prediction;
> > >
> > > (ii) 0.2% and 2.0% performance decrease on Flickr and AMiner datasets of DRAGON's final output prediction.
> > >
> > > Similarly, the previous research [2] also has the same observation that the wrong removal of this level between original nodes only causes a small robustness degradation (less than 3%) on GNNs.
> > > FNR-A (more than 25.1%) indicates the ratios of harmful malicious edges associated with attack nodes are removed by DMGAN, which is a large proportion of the total malicious edges.
> > >
> > > The above indicates that the wrong removal of true edges between original nodes by DMGAN is not a big problem to harm downstream performance, and in fact, the removal of malicious edges between original nodes and attack nodes is the key to improving the robustness of DRAGON.
> > >
> > > ---
> > >
> > >
> > > **References:**
> > >
> > > [1] Graph robustness benchmark: Benchmarking the adversarial robustness of graph machine learning, NeurIPS 2021
> > >
> > > [2] Graph structure learning for robust graph neural networks, KDD 2020

---

> > > > ### Comment · Reviewer_Fsb2 · 2023-11-22
> > > >
> > > > Thank the authors for the clarifications! My concerns about the settings are addressed. I would like to keep my rating.

---

> > > > > ### Author Response · Authors · 2023-11-22
> > > > > **Thanks for your reply!**
> > > > >
> > > > > We really appreciate your feedback, and we are happy that our response and clarifications have addressed your concerns. Thank you for recognizing our efforts on the paper!

---

### Official Review · Reviewer_qpZP · 2023-11-02

**Soundness:** 2 fair
**Presentation:** 2 fair
**Contribution:** 2 fair
**Rating:** 6
**Confidence:** 4

**Summary:**

This paper focuses on defending node injection attacks on graphs. The authors propose a pipeline called DRAGON to reconstruct a clean graph via combining denoise auto-encoder and DP-based mixture of experts. Extensive experiments are conducted to illustrate the advantage of the proposed method.

**Strengths:**

The problem is well motivated and important. The organization is clear, and the performance of the proposed method is promising.

**Weaknesses:**

1. Illustrations in some sections are not clear and easy to follow. For example in Section 4.2, the notations are not clearly introduced. Besides, the motivation of DPMoE is not well introduced. The authors should provide some intuitions/insights for leveraging Mixture-of-Experts, which I think is an important novelty of this work. But from this section, I do not get why MoE is useful though the authors empirically illustrate that in the experimental part.
2. Lemma 1 is very confusing. It is hard to understand what the authors are trying to state in this lemma and how it supports the method. The most confusing thing is that "Suppose a GNN f(·) containing DPMoE satisfies (ε,δ)-DP", how can this be supposed? I thought Lemma 1 was to prove that adding a DPMoE module to the model would make it DP, but the authors directly assume that. Besides, what does this mean "node v is robust to the features $h_v^{(l)}$"? This statement is too informal and the authors should characterize robustness in rigorous mathematical expressions.

**Questions:**

1. Why the deviation is in this form: $σ =\sqrt{p2ln(1.25/δ)/ε}$ in DPGC in section 4.2?
2. In Table 1 why adversarial training can hurt the robustness of DRAGON for most cases?

---

> ### Author Response · Authors · 2023-11-18
> **Response to Reviewer qpZP [1/3]**
>
> Dear Reviewer qpZP:
>
> We sincerely appreciate your constructive suggestions for our work. We also take your suggestions into account and address them below.
>
> ---
> **[W1. Illustrations in some sections are not clear and easy to follow. For example, in Section 4.2, the notations are not clearly introduced. Besides, the motivation of DPMoE is not well introduced. The authors should provide some intuitions/insights for leveraging Mixture-of-Experts, which I think is an important novelty of this work. But from this section, I do not get why MoE is useful though the authors empirically illustrate that in the experimental part.]**
>
> Thank you for bringing this to our attention!
>
> First, to further clarify the illustrations, we have now revised our manuscript to have clearer descriptions of notations, especially for Section 4.2 and Lemma 1.
>
> Second, the motivation for our model's DPMoE is discussed in Section 4.2 of our manuscript：
> > In comparison to the previous DP-GConv formulation in Equation (5), the proposed DPMoE layer employs a gating mechanism to control multiple experts, each equipped with multi-scale DP Gaussian noises. This allows it to effectively handle node injection attacks of varying intensities while maintaining higher anti-degraded robustness.
>
> More specifically, matching DP noise magnitudes can help model better defense attacks with different intensities, and MoE can route the node feature to the most matching DP expert to handle specific intensity attacks. The effectiveness of this design to support our motivation is empirically illustrated in the experimental part.
>
> Regarding _why MoE is useful_, previous research [6] has demonstrated that MoE architectures can adeptly manage data noise, implying an inherent potential for handling perturbations in our graph adversarial scenarios. Moreover, insights from a _theoretical study_ [5] on MoE strengthen our methodology too:
> > Each expert is good on one cluster and the router only distributes the example to a good expert. _(In Section 4 of research [5])_
> >
> > The mixture of non-linear experts can correctly recover the (underlying) cluster structure and diversify. _(In Section 6.1 of research [5])_
>
> These results do not only explain our empirical findings but also justify our intuition. That is, the MoE mechanism can recover the underlying noisy cluster structure and distribute the node to the right robust expert with matching DP noise against different attack intensities.
>
> ---

---

> ### Author Response · Authors · 2023-11-18
> **Response to Reviewer qpZP [2/3]**
>
> **[W2. Confusion about Lemma 1: "Suppose a GNN $f(·)$ containing DPMoE satisfies $(\sigma, \delta)$-DP", how can this be supposed? I thought Lemma 1 was to prove that adding a DPMoE module to the model would make it DP, but the authors directly assume that. Besides, what does this mean "node $v$ is robust to the features"? This statement is too informal and the authors should characterize robustness in rigorous mathematical expressions.]**
>
> - **Clarifying Lemma 1's purpose:** We appreciate the opportunity to elucidate Lemma 1's role. The lemma does not aim to demonstrate that the DPMoE module confers $(\epsilon, \delta)$-DP of the composition model $f(·)$; instead, it assumes $(\epsilon, \delta)$-DP property of the composition model $f(·)$ to establish an **extended robustness guarantee.** This assumption follows precedents in NLP (e.g., [7] on NAACL 2021) and CV (e.g., [8] on IEEE-S&P 2019), where language/vision models composed of the DP module can be assumed to satisfy $(\epsilon, \delta)$-DP, and then contribute to stability. The $(\epsilon, \delta)$-DP property in DPMoE is justified through its use of the classical Gaussian DP output perturbation (this mechanism is inherently $(\epsilon, \delta)$-DP [1, 2, 9]), due to the post-processing invariance of differential privacy(i.e., any computation on the output of the DP mechanism remains DP). Consequently, the output of DPMoE adheres to this differential privacy standard. For better clarity, we replace 'Suppose a GNN $f(·)$ containing DPMoE satisfies $(\sigma, \delta)$-DP' with 'For a GNN $f(\cdot)$ containing DPMoE which utilizes Gaussian DP, assume this mechanism lets the model output satisfy $(\sigma, \delta)$-DP.
>
> - **Theoretical Basis for Robustness:** As discussed in Section 4.2, Lemma 1 provides theoretical backing for the robustness of DPMoE (proof in Appendix A). This underpins our integration of DP into MoE:
>
> > By examining these inequalities, we uncover the necessary conditions for the model to maintain robustness—specifically, that the expected output for class k significantly surpasses that of any other class. The proof then extends these findings to models incorporating DPMoE, demonstrating that if a DPMoE model meets the specified conditions, its robustness is established.
>
> - **Formalizing 'Robustness to Features':** To address the ambiguity around the phrase 'node $v$ is robust to the features', we provide a rigorous definition in Appendix A, Equation 19: $\mathrm{E}(f_k(h_{v}^{'(l)})) \geq \max_{i : i \neq k} \mathrm{E}(f_i(h_{v}^{'(l)}))$. This inequality rigorously quantifies the robustness in a mathematical expression, indicating that _the expected output for the correct class $k$ remains the highest even after a node injection attack._ We use $h_{v}^{'(l)}$ to express node $v$ is attacked after aggregating injection node features.
>
> Further, we provide a thorough proof of how robustness, as stated in Lemma 1, is achieved given the precondition outlined in Equation 9. This is detailed in Appendix A of our manuscript, offering both a formal treatment and intuitive insight into the model's robustness against feature perturbations.
>
> ---
> **[Q1. Why the deviation is in this form:$\sigma = \sqrt{2\ln(1.25/\delta)}/\epsilon$ in DPGC in section 4.2?]**
>
> We follow the conclusion of $\sigma = \sqrt{2\ln(1.25/\delta)}/\epsilon$ from the classical Gaussian differential privacy mechanism [1, 2, 9]. Thank you for the reminder and we will add the clarification in the preliminary as follows:
>
> > For any $\sigma, \delta \in (0, 1)$ be two privacy parameters, the classical Gaussian differential privacy mechanism has the form $\sigma = \sqrt{2\ln(1.25/\delta)}/\epsilon$ [1, 2, 9].

---

> ### Author Response · Authors · 2023-11-18
> **Response to Reviewer qpZP [3/3]**
>
> **[Q2. In Table 1 why adversarial training can hurt the robustness of DRAGON for most cases?]**
>
> The principle behind adversarial training (AT) is to enhance model resilience against adversarial attacks. Yet, the introduction of AT does not guarantee uniform success across all contexts (which will not hurt the overall usefulness of AT across all contexts, too). Here we delve deeper into two facets:
>
> - **Potential Distribution Shift during AT**: Introducing adversarial samples via max-min adversarial training often instigates a shift in data distribution compared to real-world adversarial samples, a phenomenon discussed at length in [3]. In our findings, under non-attacked conditions or when model performance mirrors that of a non-adversarial state, the data sample distribution during AT does not align with that of the non-attacked samples. This divergence in distribution gives rise to a marginal performance dip when integrating AT with DRAGON, which mirrors observations made in previous studies [3].
>
> - **Trade-offs between Robustness and Performance**: As paper [4] illustrated, while trade-offs between robustness and performance are common in normal trained models, the trade-offs between robustness and performance on AT-trained models become more apparent when AT training is employed. In our experiments, DRAGON, when trained with AT, frequently delivers performance close to that of its non-AT-trained counterpart, sometimes even matching or outperforming it.
>
> Despite the noted trade-off, DRAGON's robustness--both with and without AT--is impressive, particularly when benchmarked against prevailing baseline methodologies. In summation, while integrating AT into DRAGON presents certain challenges, the overarching robustness and performance exhibited by our method remain noteworthy (e.g., the ability to resist the emergence of severe robustness degradation), reiterating its value in severe adversarial contexts.
>
> ---
> **References:**
>
> [1] Improving the Gaussian mechanism for differential Privacy: Analytical calibration and optimal denoising, ICML 2018
>
> [2] Calibrating noise to sensitivity in private data analysis, Theory of Cryptography 2006
>
> [3] Adversarial examples improve image recognition, CVPR 2020
>
> [4] Ensemble adversarial training: Attacks and defenses, ICLR 2018
>
> [5] Towards understanding the Mixture-of-Experts layer in deep learning, NeurIPS 2022
>
> [6] A Mixture of Experts classifier with learning based on both labelled and unlabelled data, NeurIPS 1996
>
> [7] Certified robustness to word substitution attack with differential privacy, NAACL 2021
>
> [8] Certified robustness to adversarial examples with differential privacy, IEEE Symposium on Security and Privacy (S&P) 2019
>
> [9] The algorithmic foundations of differential privacy, Foundations and Trends in Theoretical Computer Science 2014

---

> > ### Comment · Reviewer_qpZP · 2023-11-20
> >
> > Thank the authors for the detailed clarifications! My concerns are addressed. I would like to keep my rating.

---

> ### Author Response · Authors · 2023-11-21
> **Thanks for your reply!**
>
> Your feedback is greatly appreciated. We are encouraged that our response addressed your main concerns. Thank Reviewer qpZP again for taking the time to evaluate our paper!

---

### Official Review · Reviewer_owAp · 2023-11-03

**Soundness:** 3 good
**Presentation:** 3 good
**Contribution:** 2 fair
**Rating:** 6
**Confidence:** 3

**Summary:**

The authors propose a graph neural network resilient to adversarial attacks. This approach incorporates a denoising step and a novel message passing scheme that leverages a mixture of experts (trained for varying levels of noises). The superiority of the proposed method over baseline approaches is demonstrated across diverse adversarial attack scenarios.

**Strengths:**

S1. The paper is easy to follow well structured.

S2. The inclusion of a mixture of experts, each tailed for distinct noise levels, seems to be a novel and logical approach.

**Weaknesses:**

W1. The argument regarding the poor scalability of the existing method is not sufficiently convincing. For example, many extremely fast SVD algorithms have been developed for sparse graphs. It is essential for the authors to discern whether the scalability issue (e.g., O(N^2) complexity) is simply an implementation issue or it stems from fundamental limitations.

W2. It appears that the considered baselines were not specifically designed to address the attack scenarios under consideration (i.e., the injection of nodes). For instance, EvenNet is specifically designed for generalization to graphs with different degrees of homophily. Consequently, claiming superiority over them may not strongly support the effectiveness of the proposed method. The authors should consider evaluating their approach against state-of-the-art methods better aligned with the specified attack scenarios.

W3. The denoising performance of the proposed auto-encoder module is not compared with any baseline approach (e.g., Jaccard, SVD, etc).

**Questions:**

Q1. Please address W1

Q2. Please address W2

Q3. Could you provide details on how E_{r} is derived from the trained auto-encoder? Can it include new edges, or does it solely filter out some existing ones?

Q4. The results in Table 11 are not easy to understand. Can you please provide more specific details, including the numbers of TPs, TNs, FPs, and FNs?

---

> ### Author Response · Authors · 2023-11-18
> **Response to Reviewer owAp [1/3]**
>
> Dear Reviewer owAp:
>
> Thank you for your constructive questions and helpful comments. Here we address your concerns as follows. We also integrate all updates into our revised manuscript.
>
> ---
>
> **[W1&Q1. The argument regarding the poor scalability of existing methods needs to be further investigated.]**
>
> The scalability issue for the existing graph defenses has been well-discussed in previous research [1, 5, 6, 7]. We follow the discussion in previous research [5, 6, 7] on SVD to take the scalability issue as a fundamental limitation. Research [5, 7] further explains why the **SVD defense** method has a fundamental limitation on scalability and is hard to implement efficiently:
>
> >Concretely, as previous TSVD-based methods produce a dense (low-rank) adjacency matrix $A$, they involve dense matrices during GNN training, which has quadratic time/space complexity and thus cannot scale to large graphs. However, naïvely selecting the largest elements of each row in $A$ requires forming/storing $A$ first, which still has quadratic time/space complexity. _(Appendix R in research paper [7])_
>
> >GNN-SVD utilizes a low-rank approximation of the graph, that uses only the top singular components for its reconstruction... GNN-SVD and GNNGuard are not scalable to large-scale graphs due to the calculation of large dense matrices. _(Appendix A.4.3 in research paper [5])_
>
> Therefore, the SVD defense has a fundamental limitation: it produces a dense (low-rank) matrix (even for sparse graphs), and it is difficult to scale to medium/large graphs (out-of-memory on 32GB Nvidia V100 GPU) with quadratic complexity (as shown in Table 1, Flickr, Reddit, and AMiner datasets) using trivial implementation techniques.
>
> ---
> **[W2&Q2. It appears that the considered baselines were not specifically designed to address the attack scenarios under consideration (i.e., the injection of nodes). For instance, EvenNet is specifically designed for generalization to graphs with different degrees of homophily. Consequently, claiming superiority over them may not strongly support the effectiveness of the proposed method. The authors should consider evaluating their approach against state-of-the-art methods better aligned with the specified attack scenarios.]**
>
> - The baseline GAME [2] (on ICLR 2023) we used is designed to address node injection attacks. In addition, we have included recent defense methods that are widely competed under node attack scenarios, such as methods [1, 2, 3], and other competitive baselines in graph robustness benchmark (GRB) [5]. Note that GRB is a specialized, large-scale benchmark for defending against injection attacks, and we follow this benchmark to introduce several competitive baselines.
> - While we notice EvenNet [1] suffers from severe robustness degradation, we include EvenNet as baselines since its authors claim that EvenNet is still competitive against strong **graph injection attacks (GIAs)** in Appendix F of its original paper [1]:
>
> >Notwithstanding GIAs are somehow out of the scope of EvenNet, we can see that EvenNet is still competitive against strong spatial baselines in Table F, which verifies the ability of EvenNet under homophily change.
>
> - To further address your concern, we provide the performance of HANG/HANG-quad [4] (on NeurIPS 2023) and other competitive and new baselines (e.g., GAME on ICLR 2023 [2]) that are already included in our paper in Table A. In particular, HANG/HANG-quad (published 1 month after ICLR 2024 submission) is the most recent graph robust learning method against injection attacks. However, HANG suffers from scalability limitations (out-of-memory issues on large graph datasets such as Reddit and AMiner) and is less robust than DRAGON and other included baselines [2, 3].
>
> Table A. The robustness performance when defenses are under FGSM attack on the Cora dataset.
>
> |           |   nulla  |     I    |    II    |    III   |    IV    |     V    |
> |-----------|:--------:|:--------:|:--------:|:--------:|:--------:|:--------:|
> | HANG [4]      |   80.3   |   79.7   |   76.7   |   75.6   |   70.4   |   67.4   |
> | HANG-quad [4] |   78.6   |   77.7   |   75.6   |   76.8   |   68.45  |   64.23  |
> | GATGuard [3] |   81.7   |   80.6   |  _80.7_  |  _80.7_  |  _79.4_  |  _79.4_  |
> | GAME+AT [2]  | **85.5** |  _81.7_  |   80.1   |   79.1   |   78.3   |   77.3   |
> | Ours      |  _84.1_  | **84.2** | **84.1** | **84.0** | **84.0** | **83.3** |
>
>
> ---

---

> ### Author Response · Authors · 2023-11-18
> **Response to Reviewer owAp [2/3]**
>
> **[W3. The denoising performance of the proposed auto-encoder module is not compared to baselines such as Jaccard, SVD, etc.]**
>
> We have provided the performance of Jaccard, SVD, and proposed ablated DMGAN in the original paper. Table 16 in our Appendix G shows that the performance of Jaccard is less robust than GAT with AT and even close to vanilla GAT under node injection attacks. This structure-based weakness has been discussed in the benchmark [5], which avoids Jaccard as a baseline. In our manuscript, Table 1 shows that the SVD method only has close robustness to GCN with AT on Cora and is even less robust than vanilla GCN on CiteSeer. In contrast, ablated DMGAN module in Table 2 and Table 13 have better performance than these baselines (e.g., Jaccard, SVD).
>
> To further compare DMGAN with Jaccard and SVD, we conduct the experiment on them with unified backbone GCN and report the results in Table B.
>
> Table B. The robustness performance of different methods under FGSM attack on the CiteSeer dataset.
> |             | nulla    | I        | II       | III      | IV       | V        |
> |-------------|----------|----------|----------|----------|----------|----------|
> | GCN         | **71.6** | 37.9     | 33.3     | 17.8     | 16.2     | 21.0     |
> | GCN-SVD     | 68.2     | 23.4     | 22.6     | 21.7     | 19.4     | 14.6     |
> | GCN-Jaccard | 70.2     | 40.4     | 36.4     | 23.5     | 18.8     | 21.8     |
> | GCN-DMGAN   | 70.9     | **52.2** | **46.7** | **41.5** | **39.0** | **35.1** |
>
> According to our experiments, Jaccard and SVD cannot defend injection attacks well like DMGAN. They are only designed and effective to defend against structure (edge) perturbations between **original nodes** by cleaning the structure with ad hoc designs, making them less generalizable and practical to defend against advanced node injection attacks with **both** malicious features and edges. Furthermore, SVD can't scale to medium/large graphs (such as Flickr, Reddit, and AMiner datasets) since it suffers from out-of-memory on a 32GB Nvidia V100 GPU.
>
> ---
> **[Q3. Details on how $E_{r}$ is derived from the trained auto-encoder. Can it include new edges, or does it solely filter out some existing ones?]**
>
> The DMGAN, once trained, reconstructs edges by predicting their existence based on the node features within the attacked graph. This prediction aligns with link prediction methodologies. While it is true that the DMGAN may inadvertently eliminate some existing edges, the robustness of the DPMoE ensures that such occasional omissions do not adversely affect the downstream classification performance. Even at the highest level of attack intensity $V$, the proportion of existing edges mistakenly removed remains below 7.4%. We refer to the subsequent answer, which includes Table C detailing these comprehensive results.
>
> ---
> **[Q4. The results in Table 11 are not easy to understand. Can you please provide more specific details, including the numbers of TPs, TNs, FPs, and FNs?]**
>
> Thanks for bringing this to our attention.
> Let's denote TPR-N↑ (FNR-N↑) as TPR (FNR) for link prediction between normal/original nodes; and TPR-A↑ (FNR-A↑) as TPR (FNR) for link prediction between original nodes and injection/attack nodes. Since DMGAN doesn't need to predict non-existent edges on the attacked graph, we don't have FPRs and TNRs. The results in Table 11 are actually statistics of FNR-N and TPR-A. Here in Table C, we rewrite Table 11 in the following form according to your suggestion.
>
> Table C. TPRs and FNRs of DMGAN under FGSM attack on the Cora dataset.
> |        | nulla | I     | II    | III   | IV    | V     |
> |--------|-------|-------|-------|-------|-------|-------|
> | TPR-N↑ | 0.954 | 0.941 | 0.935 | 0.931 | 0.929 | 0.926 |
> | FNR-N↓ | 0.046 | 0.059 | 0.065 | 0.069 | 0.071 | 0.074 |
> | TPR-A↓ | N.A.  | 0.638 | 0.674 | 0.691 | 0.724 | 0.749 |
> | FNR-A↑ | N.A.  | 0.362 | 0.326 | 0.309 | 0.276 | 0.251 |
>
> FNR-N (less than 7.4%) indicates the ratios of edges between original nodes are wrongly removed by DMGAN. Combine with the ablation study in Table 2 and Table 13, we find that such extra wrong removal introduced by DMGAN can only lead to
>
> (i) 0.1% performance decrease on Cora and CiteeSeer datasets of DRAGON;
>
> (ii) 0.2% and 2.0% performance decrease on Flickr and AMiner datasets of DRAGON.
>
> Similarly, the previous research [8] also has the same observation that the wrong removal of this level between original nodes only causes a small robustness degradation (less than 3%) on GNNs.
> FNR-A (more than 25.1%) indicates the ratios of harmful malicious edges associated with attack nodes are removed by DMGAN, which is a large proportion of the total malicious edges.
>
> The above indicates that the wrong removal of true edges between original nodes by DMGAN is not a big problem to harm downstream performance, and in fact, the removal of malicious edges between original nodes and attack nodes is the key to improving the robustness of DRAGON.

---

> ### Author Response · Authors · 2023-11-18
> **Response to Reviewer owAp [3/3]**
>
> **References:**
>
> [1] EvenNet: Ignoring odd-hop neighbors improves robustness of graph neural networks, NeurIPS22
>
> [2] Chasing all-round graph representation robustness: Model, training, and optimization, ICLR23
>
> [3] Gnnguard: Defending graph neural networks against adversarial attacks, NeurIPS 2020
>
> [4] Adversarial robustness in graph neural networks: A hamiltonian approach, NeurIPS 2023
>
> [5] Graph robustness benchmark: Benchmarking the adversarial robustness of graph machine learning, NeurIPS 2021
>
> [6] Robustness of Graph Neural Networks at Scale, NeurIPS 2021
>
> [7] GARNET: Reduced-rank topology learning for robust and scalable graph neural Networks, Learning on Graphs Conference 2022
>
> [8] Graph structure learning for robust graph neural networks, KDD 2020

---

> > ### Comment · Reviewer_owAp · 2023-11-21
> > **Thank you**
> >
> > Thank you for the clarifications and additional efforts to address my concerns. I have raised my score.

---

> > > ### Author Response · Authors · 2023-11-21
> > > **Thanks for your feedback!**
> > >
> > > We truly appreciate your feedback. It is encouraging that our response has addressed your concerns. Thank Reviewer owAp for recognizing our efforts on the paper!

---

### Official Review · Reviewer_kprb · 2023-11-07

**Soundness:** 4 excellent
**Presentation:** 3 good
**Contribution:** 4 excellent
**Rating:** 8
**Confidence:** 4

**Summary:**

The paper develops a novel graph learning model that can handle very intense attacks and avoid heavy computation complexity. The model involves designing a new graph neural network (GNN) architecture the mixture of experts associated with differential noise. Extensive experiments are conducted to demonstrate that the proposed model can outperform many baseline methods against different attacks. Sufficient theoretical analysis is also provided.

**Strengths:**

+ It is new that the paper introduced the problem of performance degradation against intense attacks of existing models. Solving the problem is very important, especially for the data of high-risk applications. It is also interesting to consider improving computation efficiency.

+ The paper developed a new model that leverages a mixture of experts to design a new graph neural network architecture, which is novel and interesting. The theoretical analysis of the proposed method is sufficient and solid.

+ The authors have conducted sufficient experiments over multiple datasets with different intensity attacks. The proposed model outperforms many baseline methods. The improvements are significant.

**Weaknesses:**

- It seems that the current manuscript mainly focused on the node attacks. How is the performance of the proposed method against other attacks? That is, more results and discussion regarding different attacks are suggested.

- Besides a mixture of experts, there are other choices for improving GNN from an ensemble perspective. It is necessary to discuss the comparison between different methods and why the current design, i.e., MoE with differential private noise, is selected.

**Questions:**

Please see the weaknesses.

**Details Of Ethics Concerns:**

NA.

---

> ### Author Response · Authors · 2023-11-18
> **Response to Reviewer kprb**
>
> Dear Reviewer kprb:
>
> We sincerely appreciate your professional and insightful suggestions for our work. We have taken your suggestions into account and addressed them below:
>
> ---
>
> **[How is the performance of the proposed method against other attacks?]**
>
> We have also provided the analysis and the performances of our method against other graph injection attacks, graph modification attacks and graph adaptive attacks in Appendix B, E and F. In these graph attack settings, our method outperformed other popular baselines.
>
> ---
>
> **[There are other choices for improving GNN from an ensemble perspective. It is necessary to discuss the comparison.]**
>
> Besides MoE, other related ensemble methods [2, 3, 4] have been verified to improve model stability. However, recent research [1] finds simple ensemble methods face adversarial risks and have an upper bound of robustness. First, our DPMoE avoids these adversarial risks and no worst-case performance guarantees by employing the MoE with a gate network. Second, different from common ensemble methods, our method spontaneously selects the most appropriate DP expert with suitable DP noise magnitude to effectively counteract adversarial attacks at different attack intensities.
>
> ---
>
> **References:**
>
> [1] On the robustness of randomized ensembles to adversarial perturbations, ICML 2023
>
> [2] TRS: Transferability reduced ensemble via promoting gradient diversity and model smoothness, NeurIPS 2021
>
> [3] Building robust ensembles via margin boosting, ICML 2022
>
> [4] Diversifying vulnerabilities for enhanced robust generation of ensembles, NeurIPS 2020

---

### Meta-Review · Area_Chair_foS8 · 2023-12-11

**Metareview:**

Strengths:

+Important problem.
+Solid theoretical contribution, novel approach.

Weaknesses:

-Experiments could be improved in several ways pointed out by reviewers, and the authors provided additional experiments in the rebuttal. All of these should be incorporated in the final version.
-Lemma 1 and the deviation term should be appropriately introduced and discussed.

**Justification For Why Not Higher Score:**

Several reviewers remained lukewarm.

**Justification For Why Not Lower Score:**

1 Accept (8) 3 weak accepts (6)

---

### Decision · Program_Chairs · 2024-01-16

Accept (poster)